# Graph Concept Bottleneck Models

**Haotian Xu**                                                    *haotian.xu@stonybrook.edu*
*Department of Applied Mathematics & Statistics*
*State University of New York at Stony Brook*

**Tsui-Wei Weng**                                                    *lweng@ucsd.edu*
*Halıcıoğlu Data Science Institute*
*University of California, San Diego*

**Lam M. Nguyen**                                                    *lamnguyen.mltd@ibm.com*
*Thomas J. Watson Research Center*
*IBM Research*

**Tengfei Ma**                                                    *tengfei.ma@stonybrook.edu*
*Department of Biomedical Informatics*
*State University of New York at Stony Brook*

**Reviewed on OpenReview:** *https://openreview.net/forum?id=a4azUYjRhU*

## Abstract

Concept Bottleneck Models (CBMs) have emerged as a prominent framework for interpretable deep learning, providing human-understandable intermediate concepts that enable transparent reasoning and direct intervention. However, existing CBMs typically assume conditional independence among concepts given the label, overlooking the intrinsic dependencies and correlations that often exist among them. In practice, concepts are rarely isolated: modifying one concept may inherently influence others. Ignoring these relationships can lead to oversimplified representations and weaken interpretability. To address this limitation, we introduce **Graph CBMs**, a novel variant of CBMs that explicitly models the relational structure among concepts through a latent concept graph. Our approach can be seamlessly integrated into existing CBMs as a lightweight, plug-and-play module, enriching their reasoning capability without sacrificing interpretability. Experimental results on multiple real-world image classification benchmarks demonstrate that Graph CBMs (1) achieve higher predictive accuracy while revealing meaningful concept structures, (2) enable more effective and robust concept-level interventions, and (3) maintain stable performance across diverse architectures and training setups. We provide our implementation details at this URL.

## 1 Introduction

Deep neural networks have demonstrated remarkable performance and efficiency across a wide range of tasks in Computer Vision (He et al., 2016; Dosovitskiy et al., 2021; Tolstikhin et al., 2021), Natural Language Processing (Vaswani et al., 2017; Devlin et al., 2019; Brown et al., 2020; Raffel et al., 2020), and Graph Learning (Kipf & Welling, 2017; You et al., 2020b; Rong et al., 2020). Despite their success, these models are often considered black boxes due to their lack of interpretability. In safety-critical domains such as medical applications, this opacity raises concerns, as there is a growing demand for transparent and reliable decision-making processes.

To address this issue, Concept Bottleneck Models (CBMs) (Koh et al., 2020) have been introduced to enhance interpretability by mapping hidden representations to a human-understandable concept space. In CBMs, each neuron in the bottleneck layer corresponds to a specific concept with an associated confidence score. Unlike end-to-end models that map raw inputs (e.g., pixel values) directly to output labels, CBMs make

predictions based on the activation scores in the concept space. This formulation enables human intervention on the concept activation vector to correct predictions without modifying model parameters.

However, existing CBMs overlook a key aspect: *the correlations among concepts.* In reality, certain concepts often co-occur, e.g., "fur" and "whiskers" typically suggest the presence of "tail" or "paws," while "wings" and "beak" imply "feathers." Such patterns are well-established in cognitive science; for instance, Kourtzi & Connor (2011) show that visual neurons respond more strongly when related concepts appear together, suggesting the brain encodes concept relevance implicitly. Yet recent CBM approaches (Espinosa Zarlenga et al., 2022; Kim et al., 2023; Yuksekgonul et al., 2023) fail to model these natural correlations. Ignoring such relationships undermines model trustworthiness, as co-varying concepts can provide critical contextual cues to aid human understanding (Bar, 2004; Oliva & Torralba, 2007). This omission weakens interpretability and trust, especially in domains like healthcare, where co-occurring symptoms provide crucial context. For instance, during diagnosis, our model can suggest likely symptoms and their interrelations, while allowing doctors to intervene and make the final decision.

Motivated by these insights, we introduce **Graph CBMs**, a framework designed to uncover intrinsic correlations among concepts through a learnable graph structure and complement to existing CBMs with enhanced performance and interpretability. We hypothesize the existence of a unified, input-independent concept graph that encodes prior semantic knowledge, where individual input samples activate only a subset of relevant concepts. We represent this graph structure using a Graph Neural Network (GNN), and learn it via self-supervised contrastive learning. In our approach, the subgraph of activated concepts for an input image is treated as an augmented view, forming positive samples in the contrastive loss. This encourages the learned graph to capture semantically grounded relations and enables the graph to be integrated as a plug-in module across various CBM variants.

**Our key contributions are summarized as follows:**

- We propose **Graph CBMs**, which incorporate a learnable concept graph to model interactions among concepts, thereby enhancing both prediction performance and interpretability.

- We develop a self-supervised learning framework that automatically constructs latent concept graphs. Empirical results demonstrate its effectiveness on both classification and intervention (on average *1%∼2%* improvement across 8 datasets).

- Our approach shows strong generalization across settings, with or without concept annotations, and works robustly across different backbones. Moreover, Graph CBMs remain effective even under concept interventions.

## 2 Related Work

**Concept Bottleneck Models (CBMs)** improve interpretability by introducing an intermediate concept layer that predicts human-understandable concepts before performing the final linear classification (Koh et al., 2020), in which the final classification usually ignores the hidden correlation among concepts. Subsequent studies have addressed several practical limitations, such as limited concept sets and restricted predictor expressiveness (Havasi et al., 2022), and have proposed probabilistic or energy-based formulations that jointly model the input–concept–label triplet (Xu et al., 2024). Recent extensions focus on modeling richer dependencies among concepts or enabling efficient model editing: Vandenhirtz et al. (2024) capture concept dependencies via a learned multivariate normal distribution over concept logits; Hu et al. (2025) introduce influence-function–based approximations that support efficient concept or data edits without retraining; and Prasse et al. (2025) leverage foundation-model segmentations to construct concept banks for data-efficient visual concept discovery. $C^2BM$ (De Felice et al., 2025) embeds a causal graph within the concept layer, enforcing structural-causal constraints derived from prior knowledge or causal discovery. In contrast, our proposed **Graph CBMs (G-CBMs)** enable flexible, task-adaptive structure learning when causal information is limited or uncertain.

To reduce the reliance on costly concept annotations, a parallel line of work explores automatic or weakly supervised concept discovery. Label-Free CBMs (Oikarinen et al., 2023) employ large pretrained models and neuron-level interpretability tools such as CLIP-Dissect to extract candidate concepts, combined with sparse prediction layers to balance interpretability and accuracy. PCBM (Yuksekgonul et al., 2023) minimizes annotation costs by training only the final linear classifier on concept activations. Yang et al. (2023) leverage submodular optimization to select diverse and discriminative concepts, while Yan et al. (2023) further enhance concept selection under limited supervision.

A related family of methods, **Concept Embedding Models (CEMs)**, generalize CBMs by representing concepts in a continuous embedding space (Espinosa Zarlenga et al., 2022; Kim et al., 2023). While CEMs capture richer semantic relationships among concepts, they often treat these embeddings as conditionally independent and lack explicit structural modeling.

Unlike prior approaches that rely on fixed or heuristic assumptions about concept independence, our **Graph CBMs** explicitly model the latent relationships among concepts through a learned graph structure. This formulation enhances predictive accuracy and interpretability, allowing information to propagate across related concepts. As demonstrated in Tables 1 and 2, Graph CBMs can be seamlessly integrated into both supervised and label-free CBM variants, serving as a plug-in module that injects relational reasoning capabilities into the concept space.

**Graph Structure Learning (GSL)** aims to infer useful graphs from raw data. Early methods emerged in signal processing (Dong et al., 2019) and probabilistic graphical models (Yu et al., 2019). In the deep learning era, GSL is often integrated with graph neural networks (GNNs) (Franceschi et al., 2019; Kipf et al., 2018; Shang et al., 2020; Kazi et al., 2022; Ma et al., 2023). For example, NRI (Kipf et al., 2018) learns latent interaction graphs in dynamic systems using latent variables, while Franceschi et al. (2019) treats GSL as a bi-level optimization problem over discrete edge distributions. Recent advances also explore contrastive learning for hypergraphs (Wei et al., 2022). Our approach aligns with this line by learning a deterministic latent graph end-to-end through contrastive supervision, tailored to model concept relationships.

## 3 Graph Concept Bottleneck Models

Graph CBMs, as shown in Figure 1, aim to introduce latent graph structures into the concept space, enabling models to capture dependencies among concepts that are typically treated as independent in conventional CBMs. Specifically, Graph CBMs construct a *latent concept graph* where each node represents a concept and each edge encodes the learned relational strength between concept pairs. This graph is jointly learned with the concept and label predictors, allowing information to propagate among related concepts before classification. The resulting framework preserves the interpretable bottleneck of CBMs while enriching it with relational reasoning. Learning such latent topologies—often referred to as *graph structure learning*—has proven beneficial across domains such as time series forecasting (Shang et al., 2020), physical system modeling (Kipf et al., 2018), and computer vision (Kazi et al., 2022), even when explicit graph structures already exist (Franceschi et al., 2019). Similarly, in human perception, interactions among fine-grained visual concepts are known to enhance recognition and generalization (Bar, 2004; Oliva & Torralba, 2007; Kourtzi & Connor, 2011). Building on these insights, we extend the CBM framework by modeling concepts as nodes and their interactions as edges in a learned latent graph. This concept graph not only supports more expressive and accurate predictions but also enhances interpretability by revealing how concept relationships influence model decisions. In summary, Graph CBMs integrate relational inductive bias into the bottleneck layer, enabling structured concept reasoning within an interpretable deep learning paradigm.

### 3.1 Preliminaries

**Concept Bottleneck Models (CBMs)**: Given a set of images $V = v_1, v_2, \ldots, v_n$ and a predefined set of concepts $T = t_1, t_2, \ldots, t_k$, each image $v_i$ is associated with a label $y_i \in \mathcal{Y}$. A CBM consists of two components: a mapping $f_1$ from the image space to the concept score space, and a mapping $f_2$ (linear for most of CBMs) from the concept score space to the label space. The image is first projected into the concept

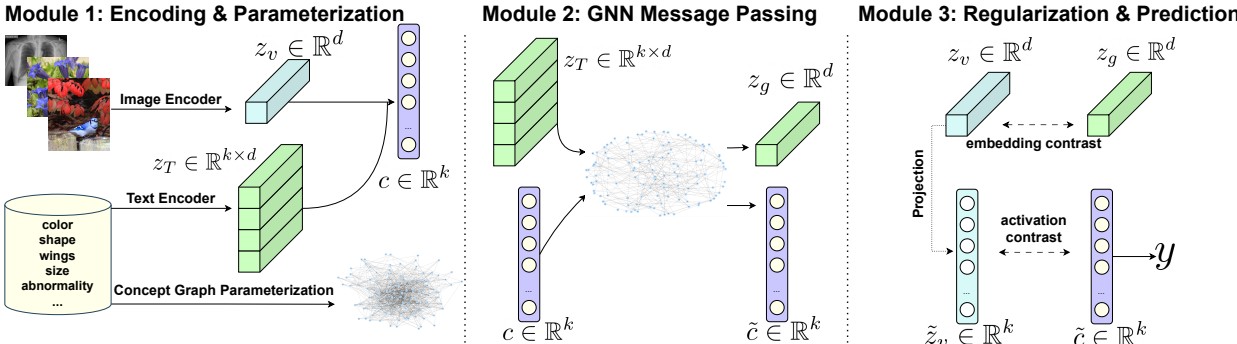

Figure 1: Overview of Graph CBM. ① Extract embeddings via pretrained encoders and initialize the graph structure. ② Update the concept embeddings and concept activations through message passing. ③ Use different granularity contrastive losses to qualify the latent graph and make final predictions.

space via $c_i = f_1(v_i; T) \in \mathbb{R}^k$, producing concept scores. These scores are used to predict the final label: $\hat{y}_i = f_2(c_i)$. Each element $c_i^j$ ($1 \leq j \leq k$) reflects the relevance between image $v_i$ and concept $t_j$.

**Our Setting**: Given an image $v_i$ and a concept set $T$, our goal is to identify a subset of activated concepts that both describe and help classify the image effectively. Following the label-free CBM setting (Oikarinen et al., 2023), we assume access to pretrained multimodal encoders: $E_v : V \rightarrow \mathbb{R}^d$ and $E_t : T \rightarrow \mathbb{R}^d$, which embed images and concepts into a shared latent space. We denote the concept embedding matrix as $z_T = E_t(T) \in \mathbb{R}^{k \times d}$. For each image $v_i$, the initial concept score vector is computed via the matrix-vector multiplication: $c_i = z_T z_{v_i} \in \mathbb{R}^k$, where $z_{v_i} = E_v(v_i)$. If ground-truth concept annotations are available, concept scores can alternatively be obtained via a learnable MLP: $c_i = \text{MLP}(z_{v_i})$.

We define the latent concept graph as $G = (V, \mathcal{A})$, where the node set $V$ corresponds to the concepts, and the adjacency matrix $\mathcal{A}$ encodes structural relations among them. In Figure 1, we show our procedures for learning the latent concept and will explain it in more detail in the following sections.

## 3.2 Module 1: Encoding and Parameterization

Module 1 prepares the inputs for our Graph-CBM model and consists of two components: (1) encoding images and concepts using pretrained encoders, and (2) parameterizing the initial latent concept graph, which will be refined in later stages.

### 3.2.1 Encoding Images and Concepts

There are two primary CBM training settings: the *label-free* and *concept-supervised* paradigms. In the label-free setting, concept annotations are unavailable, and concept sets must be defined heuristically at the dataset or instance level (Yang et al., 2023; Oikarinen et al., 2023).

In both settings, images are passed through a pretrained vision encoder to obtain image embeddings. For label-free CBMs, the concept descriptions are also processed through a pretrained language encoder to extract their embeddings. Importantly, both encoders remain necessarily frozen during CBM training to avoid converting the pipeline into a fully end-to-end model, thereby preserving the interpretability of the concept space.

### 3.2.2 Parameterizing the Initial Graph

The purpose of the latent concept graph is to model the underlying relationships among concepts. Since the concept set is defined at the dataset level, the same latent graph structure $\mathcal{A} \in \mathbb{R}^{k \times k}$, a $k \times k$ adjacency matrix over $k$ concept vetrices, is shared across all data instances. To enable finer control and interpretability, we decompose the graph node representation into two components: the semantic embedding set $V_{\text{emb}} = \{z_{T,i}\}$

and the activation vector $V_{\text{act}} = c_i$. Thus, the full graph is denoted as $G_i = (V_{\text{emb}}, V_{\text{act}}, \mathcal{A})$. $\mathcal{A}$ will be the same across instances, whereas $V_{\text{emb}}$ and $V_{\text{act}}$ will become instance-dependent during training.

$V_{\text{emb}} \in \mathbb{R}^{k \times d}$ captures the intrinsic meaning of each concept node, while $V_{\text{act}} \in \mathbb{R}^{k \times 1}$ reflects the extent to which each concept is activated per instance. The adjacency matrix $\mathcal{A}$ encodes initial structural relations between concept nodes. We initialize $\mathcal{A}$ either by computing similarities among node embeddings or randomly. Empirically, we find that random initialization achieves comparable performance, so we adopt it for simplicity. During training, $\mathcal{A}$ is optimized jointly with other model parameters and receives gradients from the task objective **CE** as well as the regularizers, following exactly the same optimization procedure as other neural parameters (e.g., MLP weights).

Although $\mathcal{A}$ is shared across instances in a dataset, the distinction brought by $V_{\text{act}}$ will trigger different nodes in the latent concept graph (we consider a concept node being activated if the corresponding entry value in $V_{\text{act}}$ is positive), resulting input-dependent graph structure in end.

### 3.3 Module 2: GNN Message Passing

With the latent concept graph, we can derive better representations of images through concept node embeddings and activations in the graph. To enhance the expressive capacity of structure learning, we adopt a multi-layer approach, where each layer is associated with its own learnable adjacency matrix. Specifically, we define the graph at layer $l$ as $G_i^l = \left(V_{\text{emb}}^{l-1}, V_{\text{act}}^{l-1}, \mathcal{A}^l\right)$, where $l$ denotes the layer index. The final latent concept graph is constructed as the union of edge sets across all layers, i.e., $\bigcup_l \mathcal{A}^l$. In practice, we use 3 layers for structure learning.

At each layer $l$, we follow a standard message passing framework (Kipf & Welling, 2017) to aggregate neighborhood information and propagate it to the concept nodes. Since concept nodes at layer $l$ are characterized by both embeddings ($V_{\text{emb}}^{l-1}$) and activations ($V_{\text{act}}^{l-1}$), we perform two distinct message passing steps at each layer.

We apply the renormalization trick introduced in (Kipf & Welling, 2017) to the adjacency matrix: $\text{Renormalize}(\mathcal{A}^l) = \tilde{D}^{-\frac{1}{2}} \tilde{\mathcal{A}}^l \tilde{D}^{-\frac{1}{2}}$, where $\tilde{\mathcal{A}}^l = \mathcal{A}^l + I$, $\tilde{D}_{ii} = \sum_j \tilde{\mathcal{A}}_{ij}^l$. Here, $I$ denotes the identity matrix, and $\tilde{D}$ is the degree matrix of $\tilde{\mathcal{A}}^l$.

The update equations for embeddings and activations at layer $l$ are given by:

$$V_{\text{emb}}^l = \sigma \left( \tilde{D}^{-\frac{1}{2}} \tilde{\mathcal{A}}^l \tilde{D}^{-\frac{1}{2}} \left[ V_{\text{act}}^{l-1} \odot V_{\text{emb}}^{l-1} \right] \right), \quad V_{\text{act}}^l = \sigma \left( \tilde{D}^{-\frac{1}{2}} \tilde{\mathcal{A}}^l \tilde{D}^{-\frac{1}{2}} V_{\text{act}}^{l-1} \right) \tag{1}$$

Here, $\odot$ denotes element-wise multiplication. The product $V_{\text{act}}^{l-1} \odot V_{\text{emb}}^{l-1}$ captures the degree to which each concept node is activated in the latent concept graph at layer $l$. $\sigma(\cdot)$ is a non-linear activation function.

### 3.4 Module 3: Regularization & Prediction

To obtain supervision on the intrinsic structure among concepts rather than on the mapping from concepts to labels, We address this by using contrastive learning, treating each image and its activated concept graph as a positive pair, since datasets do not provide ground-truth relational information between concepts. By imposing self-supervised contrastive objectives, the model is able to learn label-agnostic structures while also providing supervision in label-free settings.

After message passing, we apply a mean pooling to extract a graph-level representation for each input image: $z_{g_i} = \text{MeanPooling}(V_{\text{emb}}^m) \in \mathbb{R}^d$, where $m$ is the number of layers. As previously discussed, the concept graph acts as an augmentation technique for input images. Under the assumption of intrinsic concept relationships, we introduce a contrastive regularization term based on the normalized temperature-scaled cross-entropy loss (NT-Xent) (Chen et al., 2020). Moreover, relying solely on the embedding-wise contrastive loss yields suboptimal performance (see Tables 3 and 4). Inspired by Ribeiro et al. (2017); Sun et al. (2019); You et al. (2020a); Wang et al. (2022), we design a second contrastive loss at a different granularity to further regularize the updated concept scores $\tilde{c}_i$. We treat $\tilde{c}_i = V_{\text{act}}^m$ as the positive pair for image $v_i$ in $\mathbb{R}^k$ (the concept space). We project $z_{v_i}$ into the same space using an MLP layer $f_3 : \mathbb{R}^d \to \mathbb{R}^k$, yielding $\tilde{z}_{v_i} = f_3(z_{v_i}; \phi)$ as the anchor

point in the concept space.

$$\mathcal{L}_{emb} = -\log\left(\sum_{i=1}^{n} \frac{e^{\text{sim}(z_{v_i}, z_{g_i})/\tau}}{\sum_{j=1, j\neq i}^{n} e^{\text{sim}(z_{v_i}, z_{g_j})/\tau}}\right), \quad \mathcal{L}_{act} = -\log\left(\sum_{i=1}^{n} \frac{e^{\text{sim}(\tilde{z}_{v_i}, \tilde{c}_i)/\tau}}{\sum_{j=1, j\neq i}^{n} e^{\text{sim}(\tilde{z}_{v_i}, \tilde{c}_j)/\tau}}\right), \tag{2}$$

where $\text{sim}(\cdot, \cdot)$ denotes the cosine similarity, and $\tau$ is a temperature hyperparameter (set to $\tau = 0.3$ in our experiments). In each mini-batch, the positive pair consists of the image embedding $z_{v_i}$ (or $\tilde{z}_{v_i}$) and its corresponding concept graph embedding $z_{g_i}$ (or $\tilde{c}_i$), while the negative pairs are the graph embeddings of all other images.

Specifically, the embedding-level contrastive loss $\mathcal{L}_{emb}$ is computed in the latent representation space $\mathbb{R}^d$, while the activation-level contrastive loss $\mathcal{L}_{act}$ is computed in the concept activation space. $\mathbb{R}^K$, after the mapping $f_3 : \mathbb{R}^d \rightarrow \mathbb{R}^K$. Thus, the two contrastive losses operate at different layers and regulate the latent graph structure from complementary perspectives. $\mathcal{L}_{emb}$ enforces alignment and uniformity between the image and graph-level embeddings, while $\mathcal{L}_{act}$ encourages concept activation vectors to serve as effective hidden representations of the input images (Wang & Isola, 2020).

In addition, we hypothesize that the performance degradation observed when using CBMs is partly due to information loss during the projection from the high-dimensional image latent space to a lower-dimensional concept space. This is supported by the fact that directly classifying the image latent features (with only a single trainable linear prediction head) achieves 81.19% accuracy on CUB, whereas the intervening concept bottleneck reduces the usable signal. Consequently, making the concept space more faithful to or better aligned with the geometry of the image feature space can benefit downstream classification. This motivation underlies the second contrastive loss, and our empirical results in Section 4 corroborate this intuition.

Combining both with a prediction loss and a sparsity regularization term, the final loss becomes:

$$\mathcal{L} = \text{CE}(\hat{y}_i, y_i) + \alpha(\mathcal{L}_{emb} + \mathcal{L}_{act}) + \beta\,\ell_1(\mathcal{A}), \tag{3}$$

where $\text{CE}(\cdot, \cdot)$ is the cross-entropy loss, $\ell_1(\cdot)$ denotes L1 regularization, and $\alpha$, $\beta$ are weighting hyperparameters. The prediction $\hat{y}_i$ is computed by passing $\tilde{c}_i$ through another MLP $f_2 : \mathbb{R}^k \rightarrow \mathcal{Y}$, i.e., $\hat{y}_i = f_2(\tilde{c}_i)$.

The above formulation is primarily designed for the *label-free* setting. When concept supervision is available, the model directly learns concept representations from ground-truth annotations. In such cases, we no longer require a language encoder to obtain concept embeddings, and the contrastive loss reduces to $\mathcal{L}_{act}$, alongside a sparsity regularization term. The final loss function in the concept-supervised setting becomes:

$$\mathcal{L} = \text{CE}(\hat{y}_i, y_i) + \text{BCE}(\hat{c}_i, \tilde{c}_i) + \alpha\mathcal{L}_{act} + \beta\,\ell_1(\mathcal{A}), \tag{4}$$

where $\text{BCE}(\cdot, \cdot)$ is the binary cross-entropy loss. Since concept annotations provide structural cues, the BCE loss guides the latent graph to align with these ground-truth relationships. Simultaneously, $\mathcal{L}_{act}$ regularizes the graph by treating $\tilde{c}_i$ as a meaningful latent representation of image $v_i$. Together, they enable the model to learn a high-quality latent graph structure.

| Base experiments | CUB | Flower102 | HAM10000 | Cifar-10 | Cifar100 |
|---|---|---|---|---|---|
| LF-CBM | 73.90% (±0.28%) | 84.77% (±0.59%) | 66.76% (±0.43%) | 86.40% (±0.10%) | 65.16% (±0.14%) |
| Graph-(LF-CBM) | **75.59%** (±0.18%) | **86.00%** (±0.98%) | **67.47%** (±0.61%) | **86.54%** (±0.15%) | **65.96%** (±0.16%) |
| PCBM | 74.69 % (±0.16%) | 79.01% (±1.19%) | 77.61% (±0.60%) | 95.71% (±0.07%) | 80.02% (±0.39%) |
| Graph-PCBM | **77.95%** (±0.69%) | **89.25%** (±0.69%) | **78.50%** (±0.52%) | **95.95%** (±0.09%) | **80.86%** (±0.26%) |

Table 1: **Graph CBM can better capture image information.** We report the mean and standard deviation from 10 random runs.

# 4 Experiments & Results

## 4.1 Setup

**Datasets**: For evaluating Graph CBMs, we choose various real-world datasets ranging from common objects to dermoscopic images. We introduce 1) common objects: CUB (Wah et al., 2011), Flower102 (Nilsback & Zisserman, 2008), AwA2 (Xian et al., 2018), and CIFAR (Krizhevsky et al., 2009) ; and 2) medicial domains: HAM10000 (Tschandl, 2018) and ChestXpert (Irvin et al., 2019). Dataset details can be found in appendix A.

Since many datasets do not contain concept annotations, we will use LLMs to generate and filter the concept set for each dataset. In CUB and CIFAR datasets, we use the same concept sets provided by (Oikarinen et al., 2023). Furthermore, we found that concepts can be redundant and noisy in CIFAR datasets; thus, we apply the submodular filtering algorithm (Yang et al., 2023) to reduce the number of concepts in (G-)PCBM experiments. For Flower and HAM10000, we use the concept candidates in (Yang et al., 2023).

**Baseline**: We choose the state-of-the-art CBM models that do not need concept annotations during training, i.e., LF-CBM (Oikarinen et al., 2023) and Post-hoc CBMs (PCBM) (Yuksekgonul et al., 2023). For multi-modality encoders, we choose the standard CLIP (Radford et al., 2021). For a fair comparison with Label-free CBMs, we use CLIP(ViT-B/16) and CLIP(RN50) as the image encoder and the backbone (ResNet-18 trained on CUB from imgclsmob as the backbone for CUB). In the concept-supervised setting, standard CBMs (Koh et al., 2020) and CEMs (Espinosa Zarlenga et al., 2022) serve as our baseline, and we still use CLIP(ViT-B/16) to extract features on CUB (112 concepts) and AwA2 (85 concepts) dataset. For ChestXpert (11 concepts), we use BioViL (Bannur et al., 2023) as the image encoder, as BioViL is pre-trained on large X-ray datasets. We use the Adam optimizer and cosine scheduler during training. More configuration details and efficiency concerns are offered in Appendix B.

## 4.2 Latent Graphs Enhance Model Performance

**Graph-LF-CBM and Graph-PCBM can substantially outperform their counterparts without graphs under label-free settings.** Observing the solid enhancement in Table 1 of having a latent concept graph across different datasets, we can conclude the effectiveness of our proposed method. To better understand the benefit of having a latent concept graph, we draw T-SNE plots for $c_i$ ($\tilde{c}_i$) in Figure 6 Appendix E, showing that models with latent concept graphs can better cluster concept activations to corresponding label groups. In Table 10 Appendix G, our method surpasses the baseline on large-scale datasets, which further validates the effectiveness of using latent concept graphs.

**Graph CBMs can generalize their improvements to concept-supervised settings.** In Table 2, adding the latent graph can increase the label prediction performance for all the datasets from different domains, and match or even surpass the baseline's capability on concept prediction. It is important to note that the latent graph is primarily designed to **leverage interactions among concepts** to improve label prediction and enable more effective interventions, rather than directly optimize concept-level predictions. Furthermore, our models still demonstrate notable improvements in some cases, particularly, G-CEM not only boosts label prediction accuracy over standard CEM but also narrows the gap in concept prediction performance compared to CBM-based approaches.

## 4.3 Comparison with SOTAs

In Figure 2, we compare our proposed method with state-of-the-art models to demonstrate the effectiveness of constructing a latent concept graph. We evaluate our method under both *label-free* and *concept-supervised* settings using the CUB dataset, a common benchmark for concept-based modeling. For the label-free setting, we include additional strong baselines such as BotCL (Wang et al., 2023), CDM (Panousis et al., 2023), LaBo (Yang et al., 2023), and Res-CBM (Shang et al., 2024). For the concept-supervised setting, we compare against ProbCBM (Kim et al., 2023), HardAR (Havasi et al., 2022), E-CBM (Xu et al., 2024), and S-CBM (Vandenhirtz et al., 2024).

| Method | CUB | | AwA2 | | ChestXpert | |
|---|---|---|---|---|---|---|
| | Label | Concept | Label | Concept | Label | Concept |
| CBM | 78.45% (±0.32%) | **70.40%** (±0.09%) | 95.24% (±0.07%) | **97.48%** (±0.05%) | 66.40% (±0.86%) | **83.41%** (±0.44%) |
| Graph-CBM | **80.03%** (±0.16%) | 68.33% (±0.21%) | **95.34%** (±0.08%) | 97.05% (±0.07%) | **66.82%** (±0.47%) | 83.20% (±1.02%) |
| CEM | 80.86% (±0.12%) | 61.34% (±0.21%) | 95.21% (±0.46%) | 98.16% (±0.30%) | 66.73% (±0.53%) | 77.93% (±0.43%) |
| Graph-CEM | **81.11%** (±0.26%) | **61.53%** (±0.22%) | **95.49%** (±0.43%) | **98.62%** (±0.13%) | **66.93%** (±0.62%) | **78.27%** (±0.13%) |

Table 2: Comparison between label prediction and concept prediction. We report the average accuracy (for label prediction) and roc-auc (for concept prediction) from 10 random seed experiments.

Under the label-free setting, LaBo and Res-CBM perform well but rely on large concept sets (e.g., 10,000 for CUB). In contrast, our G-PCBM achieves higher accuracy using only 200 concepts, fewer training epochs, and a comparable backbone (e.g., ViT-L/14 improves G-PCBM from 77.1% to 82.3%). For other reference, (Yan et al., 2023) reports 63.9% with the same setup. This demonstrates the efficiency and effectiveness of leveraging latent graphs in CBMs.

In the concept-supervised setting, our models, G-CBM and G-CEM, perform competitively with the best current methods. HardAR and E-CBM both aim to model latent concept correlations but follow fundamentally different approaches. HardAR focuses on autoregressively generating concept scores, assuming sequential dependencies. Our method, by contrast, employs graph-based modeling, which is inherently **permutation-invariant** Maron et al. (2019); Bronstein et al. (2021) and better suited to capturing rich, non-linear concept interactions. Furthermore, our latent graphs can seamlessly plug into HardAR's framework to further enhance its relational reasoning through message passing.

Compared to E-CBM, which encodes concept dependencies implicitly via energy-based modeling, our approach offers **explicit** graph structures that are easier to visualize and interpret. This transparency is especially valuable for downstream tasks such as intervention (see Section 4.4). The learned graph not only improves model performance during training but also enables more effective post-hoc analysis and decision manipulation.

Importantly, our method is **orthogonal** to existing CBM designs and can be **flexibly integrated** into a variety of architectures. To support this claim, we present a case study with CDM (Panousis et al., 2023) in Appendix H, showing how latent graphs can complement other CBM variants. Moreover, unlike models such as HardAR, E-CBM, or S-CBM that require datasets with explicit concept annotations, our latent graph framework is **versatile** and can be applied in both concept-supervised and label-free CBM settings, broadening its applicability across domains.

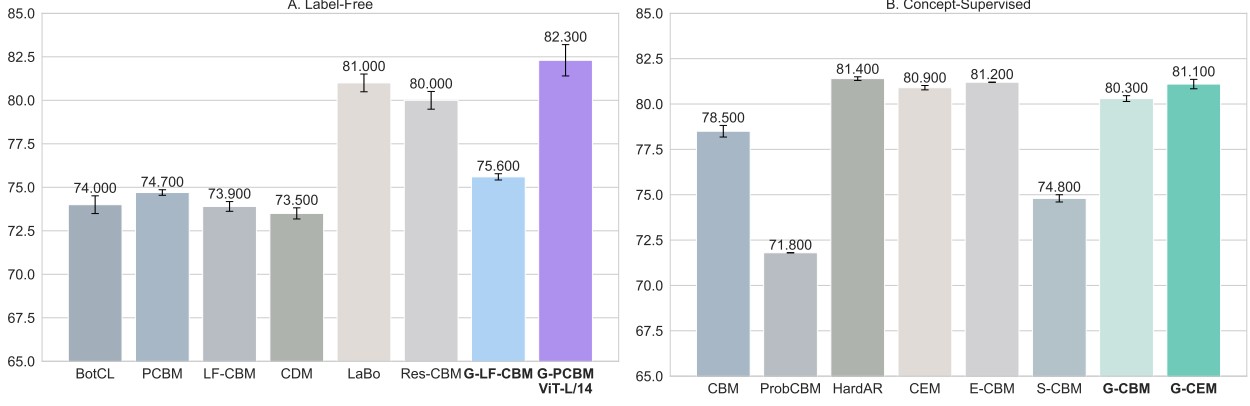

Figure 2: We compare our models with current SOTA results for corresponding training and dataset settings, i.e., label-free and concept-supervised. We report label accuracy in both subfigures.

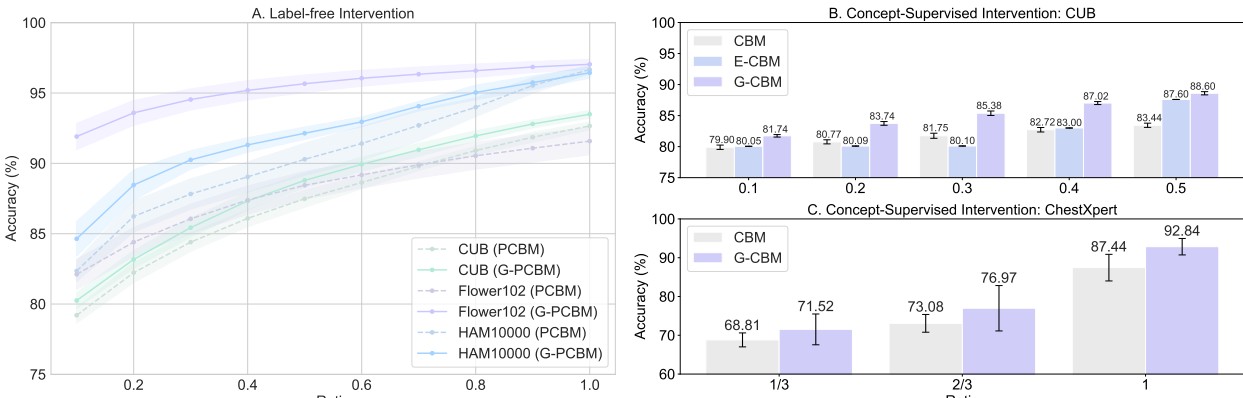

Figure 3: This figure shows the effectiveness of using latent graphs when intervening concepts. Subfigure A compares the after-intervention performance between PCBM and G-PCBM across three datasets under the label-free setting; subfigures B and C select concept-supervised datasets. Full ratio intervention comparisons between G-CBM and E-CBM are in appendix J.

## 4.4 Intervention Dynamics

A core advantage of CBMs lies in their **intervenability**: allowing users to adjust concept activations to correct false predictions and improve trust. Beyond enhancing label prediction post-training, we evaluate whether latent graphs further strengthen CBMs during concept interventions. *Notably, the benefit of incorporating latent graphs becomes more pronounced in this setting, as relational cues help guide more effective interventions.* Following the UCP policy (Shin et al., 2023), we select relevant concepts and apply a simple *Lazy Intervention* strategy (Appendix D) to modify activation values.

**In both label-free and concept-supervised settings, incorporating a latent graph improves intervention performance.** As defined in Equation 1, the message passing mechanism allows changes in the intervened activation vector $c_i$ to propagate to neighboring concepts via the latent graph, amplifying the effect to calculate the final $\tilde{c}_i$. Thus, models with graphs implicitly intervene more broadly, leading to more robust corrections, as shown in Figure 3 A.

In the concept-supervised setting (Figure 3 B & C), we compare performance on the CUB and ChestXpert datasets. Unlike E-CBM (Xu et al., 2024), which requires iterative energy calculations and gradients for intervention, G-CBM achieves strong results via a single forward pass. This efficiency, combined with high accuracy, highlights the benefit of explicitly modeling concept relations. We hypothesize that concept supervision encourages meaningful correlations, which the latent graph further captures and utilizes during intervention. The improved performance across datasets supports the positive impact of latent structure in enhancing interpretability and trust.

## 4.5 How Contrastive Terms Affect Model Performance

**Using both $\mathcal{L}_{emb}$ and $\mathcal{L}_{act}$ is crucial for Graph LF-CBM.** Table 3 examines different combinations of contrastive losses on the CUB dataset. We find that using only $\mathcal{L}_{emb}$ marginally improves performance, while incorporating $\mathcal{L}_{act}$ leads to further gains. However, on HAM10000 and CIFAR-100, applying either loss alone fails to outperform the LF-CBM baseline. Notably, using both $\mathcal{L}_{emb}$ and $\mathcal{L}_{act}$ consistently achieves the best accuracy across tasks. When only $\mathcal{L}_{emb}$ is used, the learned graph becomes overly dense due to insensitivity to the $\ell_1$ regularization on edge weights. Conversely, relying solely on $\mathcal{L}_{act}$ tends to yield overly sparse graphs. Thus, combining both objectives enables effective control of graph complexity and improves representation quality.

**With target supervision, contrastive regularization further enhances the expressivity of Graph PCBM.** Contrastive learning provides a self-supervised signal by treating concepts as alternative views of the same input. We investigate whether contrastive regularization still benefits Graph PCBM under target supervision. As shown in Table 4, applying both $\mathcal{L}_{emb}$ and $\mathcal{L}_{act}$ improves performance across CUB,

| Regularizer | Existence | | | |
|---|---|---|---|---|
| $\mathcal{L}_{emb}$ | - | - | ✓ | ✓ |
| $\mathcal{L}_{act}$ | - | ✓ | - | ✓ |
| Dataset | Performance | | | |
| CUB | 73.90% (±0.65%) | 75.38% (±0.12%) | 74.11% (±0.17%) | **75.59%** (±0.18%) |
| HAM10000 | 66.76% (±0.41%) | 62.87% (±1.34%) | 65.10% (±0.47%) | **67.47%** (±0.61%) |
| Flower102 | 84.77% (±0.44%) | 85.99% (±0.68%) | 76.78% (±0.49%) | **86.00%** (±0.80%) |
| CIFAR10 | 80.76% (±0.17%) | 72.42% (±1.21%) | 84.08% (±0.22%) | **84.65%** (±0.10%) |
| CIFAR100 | 65.16% (±0.14%) | 59.49% (±0.34%) | 62.97% (±0.21%) | **65.96%** (±0.17%) |

Table 3: Multi-level contrastive learning is crucial for self-supervised concept graph qualities. If we train Graph LF-CBMs with one excluding contrast loss, the model can fail to yield a good latent structure; while considering both contrast losses, models can gain more benefits.

| Method (Graph PCBM) | CUB | HAM10000 | CIFAR-100 |
|---|---|---|---|
| w/o ($\mathcal{L}_{emb} + \mathcal{L}_{act}$) | 76.89% (±0.65%) | 77.66% (±0.41%) | 75.67% (±0.35%) |
| w ($\mathcal{L}_{emb} + \mathcal{L}_{act}$) | **77.95%** (±0.69%) | **78.50%** (±0.52%) | **80.86%** (±0.26%) |

Table 4: Adding contrast loss to target supervision can help model performance. Having the two contrast regularizers will lead to higher accuracy for label prediction.

HAM10000, and CIFAR. On CIFAR-100 in particular, we observe a performance gain of over 5%. Without contrast losses, the model relies entirely on target supervision and $\ell_1$ regularization, making the learned graph highly sensitive to task-specific supervision signals and regularization strength. Contrast regularization thus provides a stable, auxiliary training signal that improves generalization and structure learning.

To further investigate the underlying source of effectiveness in the proposed methodology, we conduct an additional ablation study on the CUB dataset under the label-free setting. The goal of this experiment is to disentangle whether the observed improvement is primarily driven by (i) the use of contrastive learning or (ii) the introduction of the GNN message-passing module. Specifically, we compare: (1) the effect of adding $\mathcal{L}_{emb}$ and/or $\mathcal{L}_{act}$ to PCBM, and (2) the same set of losses applied to Graph-PCBM. The results are presented in Figure 4.

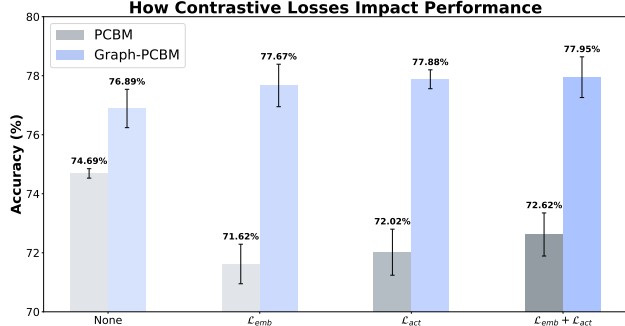

Figure 4: Contrastive learning shows positive impact mainly when the idea of graph is implemented.

From this comparison, we observe that the dominant source of improvement arises from enabling message passing over concepts. Classical CBMs treat concepts as independent units, whereas Graph-CBM allows concepts to interact through a learned graph. We argue that this aligns better with the semantic structure of many domains. For example, in fine-grained recognition tasks, certain concepts reinforce or suppress others, and modeling such dependencies improves both predictive performance and intervention.

To summarize the analyses in this section, we emphasize that **model capacity is not the main driver of improvement**. The GNN introduces only $O(k^2)$ additional parameters, does not fine-tune or alter the image encoder, and the learned adjacency matrices are sparse. Therefore, the performance gain cannot be

attributed to increased representational capacity. Instead, the benefit appears to stem primarily from adopting GNN-based message passing rather than from contrastive learning alone. This reinforces our hypothesis that modeling intrinsic concept relations via a latent graph is beneficial.

## 4.6 How to Interpret The Concept Graph

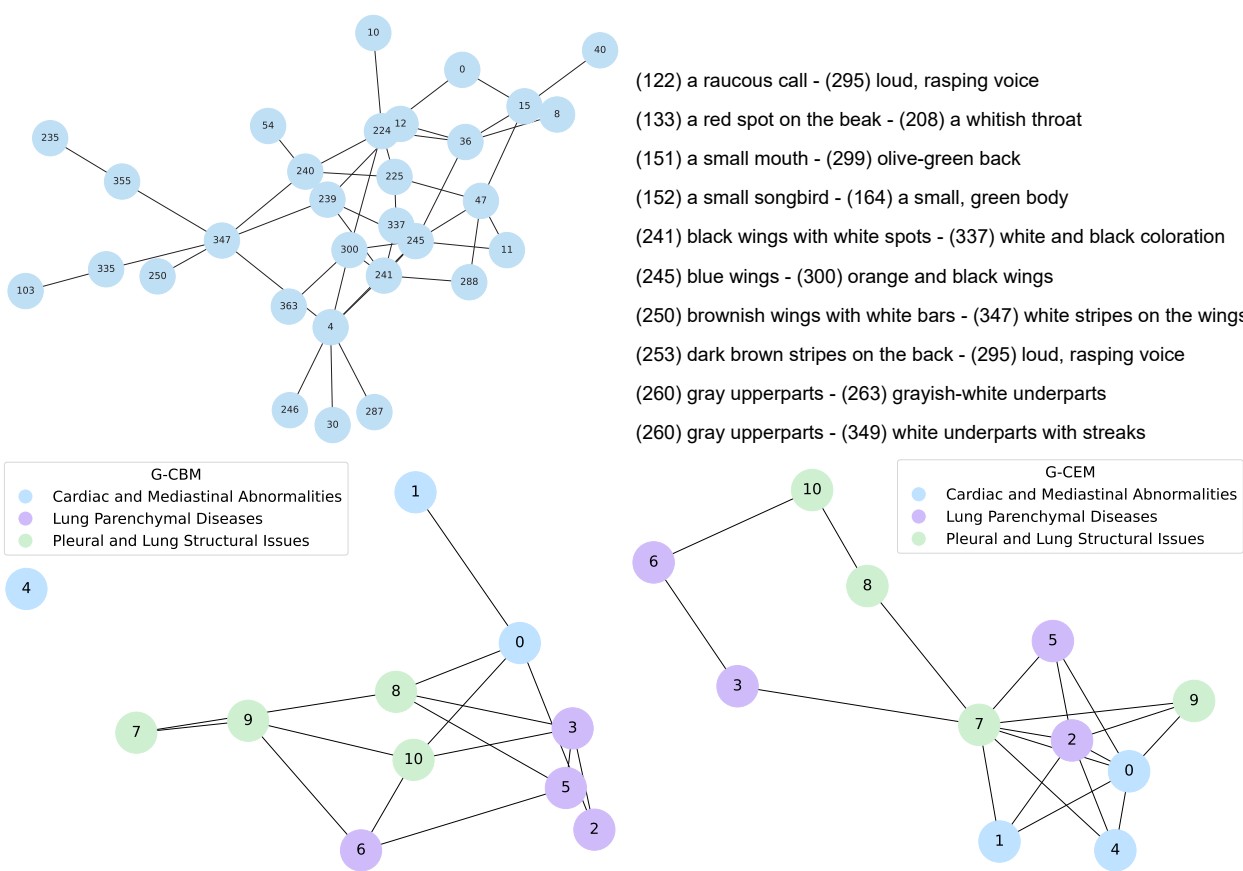

Figure 5: *Top*: Illustrating part of the learned latent graph under the label-free setting on CUB, alongside representative connected concept pairs. *Bottom*: G-CBM and G-CEM capture similar and trustworthy concept interactions for the ChestXpert dataset.

**Latent graph can perform comparably to or recover a ground-truth concept graph in both settings.** We use ChatGPT (Ouyang et al., 2022) to generate 50 highly correlated concept pairs to build a "real-world" concept graph for *label-free* setting. Substituting the learned graph with this LLM-derived structure yields a CUB accuracy of **77.69%**, while our learnable graph achieves **77.14%**, demonstrating similar efficacy. Interestingly, LLMs excel at capturing surface-level similarities (e.g., color or body parts), while our method also discovers non-obvious correlations (e.g., ''a loud, harsh cry'' – ''a raucous call''). In Figure 5, we also observe that concepts sharing the same label tend to be connected, regardless of architecture, validating that learned graphs reflect semantically meaningful groupings. In Appendix I, we can further extract salient subgraphs with fewer edges that maintain (or even improve) label accuracy and interpretability. Figures 10 & 11 show that key concept relations are retained, reinforcing the utility of our latent graph formulation. Interestingly, **latent graphs behave distinctly to different training regimes.** As shown in Figure 13, label-free models learn sparser graphs, while concept-supervised ones favor denser connections. More analyses are in Appendix K.

**Latent graphs improve robustness under concept attacks.** We conduct an experiment on the CUB dataset in Appendix F where we randomly mask or perturb a fixed number of concepts in three different

groups: (1) connected concepts, (2) isolated concepts, and (3) random selections from the entire concept set. As shown in Figure 7, corrupting connected concepts results in significantly less performance degradation compared to isolated or randomly selected ones. This suggests that the latent graph enables the model to recover corrupted concept information by aggregating signals from clean neighboring nodes, thereby enhancing resilience against noise or missing data.

## 5 Limitations & Conclusions

Graph CBMs currently capture only latent concept interactions, without modeling hierarchies or complex relationships. Incorporating external knowledge about concept hierarchy could enhance interpretability, and future work will explore hierarchical structures beyond pairwise interactions.

In this paper, we introduced Graph CBMs, a simple yet effective framework that incorporates graph structures to model concept correlations and enhance interpretability. Our method is orthogonal to existing CBMs and can be seamlessly integrated to boost performance while supporting targeted concept interventions. Notably, the learned latent graphs can match the effectiveness of ground-truth concept graphs. Future directions include leveraging external knowledge graphs for graph distribution modeling and extending latent graph techniques to interpret individual neurons within CBMs (Oikarinen & Weng, 2024).

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

# Appendix

## Table of Contents

## A   Dataset Introduction

- Caltech-UCSD Birds-200-2011, CUB (Wah et al., 2011): CUB is the most widely-used dataset for fine-grained visual categorization tasks. It contains 11,788 images of 200 subcategories belonging to birds, we follow the same data processing as done in the Label-free-CBM (Oikarinen et al., 2023) setting to select 5,990 images for the training set and 5,790 images for the validation set. For the concept supervised setting, we process the same way as (Koh et al., 2020) and (Espinosa Zarlenga et al., 2022), selecting 112 attributes as the concepts and use the same data splits.

- Oxford 102 Flower (Nilsback & Zisserman, 2008): is an image classification dataset comprising 102 flower categories. The flowers chosen to be flowers commonly occur in the United Kingdom. Each class consists of between 40 and 258 images. We follow the torchvision dataset setting (maintainers & contributors, 2016) and only use the training set for training and the validation set for testing.

- CIFAR-10 & CIFAR-100 (Krizhevsky et al., 2009): The CIFAR-10/100 dataset (Canadian Institute for Advanced Research, 10 classes) is a subset of the Tiny Images dataset and consists of 60,000 32x32 color images. CIFAR-10 labels images with one of 10 mutually exclusive classes, and CIFAR-100 has 100 classes grouped into 20 superclasses. There are 6000 images per class with 5000 training and 1000 testing images per class in CIFAR-10, while CIFAR-100 divides the dataset into 500 training images and 100 testing images per class.

- HAM10000 (Tschandl, 2018): A dataset of 10000 training images for detecting pigmented skin lesions. It contains 7 labels and a representative collection of all important diagnostic categories in pigmented lesions HAM10000 provides 10,000 training images and 1,511 testing images.

- AwA2 (Xian et al., 2018) is a zero-shot learning dataset containing 37,322 images and 50 animal classes. Unlike the CUB dataset (Wah et al., 2011) in which concepts are defined at the instance level, images under the same label inside AwA2 (Xian et al., 2018) will share the same concepts. We use all 85 attributes as concepts.

- ChestXpert (Irvin et al., 2019) is a large dataset of chest X-rays and competition for automated chest X-ray interpretation, consisting of 224,316 chest radiographs of 65,240 patients, which features 14 uncertainty observations and radiologist-labeled reference standard evaluation sets. ChestXpert (Irvin et al., 2019) does not provide binary label classification, so we cluster 11 out of 14 observations into 3 categories. The detailed data processing can be found in appendix C.

The number of concepts for datasets is as follows: 370 for CUB, 108 for Flower, 48 for HAM10000, 30 for CIFAR-10, and 50 for CIFAR-100. For (G-)LF-CBM experiments, we use the original number concepts without any filtering (143 for CIFAR-10 & 892 for CIFAR-100).

## B    Configuration and Running Environments

| dataset | training epochs | Learning Rate | $\alpha$ | $\beta$ |
|---------|----------------|---------------|----------|---------|
| CUB | 100 | 1e-3 | 0.1 | 0.2 |
| Flower102 | 500 | 1e-3 | 0.1 | 0.05 |
| HAM10000 | 100 | 1e-3 | 0.1 | 0.05 |
| CIFAR-10 | 100 | 1e-3 | 0.1 | 0.1 |
| CIFAR-100 | 100 | 1e-3 | 0.1 | 0.05 |

Table 5: Training Configuration for Graph LF-CBM

| dataset | training epochs | Learning Rate | $\alpha$ | $\beta$ |
|---------|----------------|---------------|----------|---------|
| CUB | 100 | 1e-3 | 0.1 | 0.2 |
| Flower102 | 100 | 1e-3 | 0.1 | 0.01 |
| HAM10000 | 100 | 1e-3 | 0.1 | 0.05 |
| CIFAR-10 | 50 | 1e-3 | 0.1 | 0.05 |
| CIFAR-100 | 30 | 1e-3 | 0.1 | 0.05 |

Table 6: Training Configuration for Graph PCBM

| dataset | training epochs | Learning Rate | $\alpha$ | $\beta$ |
|---------|----------------|---------------|----------|---------|
| CUB | 100 | 1e-3 | 0.05 | 0.0 |
| AwA2 | 50 | 1e-3 | 0.01 | 0.0 |
| ChestXpert | 50 | 1e-3 | 0.03 | 0.0 |

Table 7: Training Configuration for Graph CBM

| dataset | training epochs | Learning Rate | $\alpha$ | $\beta$ |
|---------|----------------|---------------|----------|---------|
| CUB | 100 | 1e-3 | 0.07 | 0.0 |
| AwA2 | 50 | 1e-3 | 0.07 | 0.0 |
| ChestXpert | 50 | 1e-3 | 0.07 | 0.0 |

Table 8: Training Configuration for Graph CEM

We run all the experiments on a single GPU (NVIDIA A100). The GPU memory for Graph LF-CBMs and Graph PCBMs is less than 10GB with a batch size of 512 for all datasets. The full training run takes from 3 minutes to 2.5 hours depending on the dataset size and the number of training epochs. In practice, Graph LF-CBMs trained on CIFAR-100 take 2.5 hours, while Graph PCBMs trained on Flower need less than 3

minutes for execution. Graph CBMs and Graph CEMs on average take 8∼12 mins to finish training on the CUB and ChestXpert datasets, while they spend about 3∼5 mins on the AWA2 dataset. We also offer time measurements in Table 9: since the latent graph introduces more computational units, the model will be unavoidable to spend more time on training.

| Model | Training | Inference |
|-------|----------|-----------|
| PCBM | 5.7it/s | 2.09it/s |
| G-PCBM | 9.43it/s | 2.12it/s |
| CBM | 6.12it/s | 2.90it/s |
| G-CBM | 7.89it/s | 2.57it/s |

Table 9: Time Measurement between our proposals and baselines

## C   Data Processing for ChestXpert

CheXpert (Irvin et al., 2019) uses "No Finding" to indicate the abnormality of patients' chest radiographs, and it is highly unbalanced. We will then cluster CheXpert's concepts into 3 different labels, and we will use the frontal and lateral X-ray images for each patient. The concepts and labels are classified in this way:

- **Group 1: Cardiac and Mediastinal Abnormalities**:
    - Enlarged Cardiomediastinum
    - Cardiomegaly
    - Edema (related to heart conditions)

- **Group 2: Lung Parenchymal Diseases**:
    - Lung Opacity
    - Lung Lesion
    - Consolidation
    - Pneumonia

- **Group 3: Pleural and Lung Structural Issues**:
    - Atelectasis (collapse of lung tissue)
    - Pneumothorax (air in pleural space)
    - Pleural Effusion
    - Pleural Other

If one patient meets multiple abnormal conditions (multi-labeled), we will select the most significant abnormality. We choose not to include the *Fracture* observation, as it forms a cluster itself and shares no commonalities with other observations.

## D   Lazy Intervention

**Lazy Intervention**: We define two sets of concept scores

$$\mathcal{R} = \{c_i \mid h(\tilde{c}_i) = y_i\}, \quad \mathcal{W} = \{c_i \mid h(\tilde{c}_i) \neq y_i\},$$

where $\mathcal{R}$ and $\mathcal{W}$ are sets of concept scores making right and wrong predictions. We can further partition them using class labels, so $\mathcal{R} = \cup_{i=1}^{m} \mathcal{R}^i$, and so does $\mathcal{W}$. We then define the difference set,

$$\mathcal{D} = \{\text{mean}(\mathcal{R}^j) - \text{mean}(\mathcal{W}^j) \mid 1 \leq j \leq m\},$$

$m$ is the number of classes. Each $d^j \in \mathcal{D}$ can be viewed as a prototype of the intervention vector for $j$-th class. The intervention procedure will be

$$\text{Intervention} = \{c_i + d^j \mid \forall c_i \in \mathcal{W}^j, 1 \le j \le m\}.$$

When the dataset contains concept annotations, the $\mathcal{R}$ records positively classified concepts, and the $\mathcal{W}$ records falsely classified concepts. At the intervention step, *Lazy intervention* will only intervene on falsely classified concepts.

## E   Having a Latent Concept Graph Enriches concept activations Expressivity

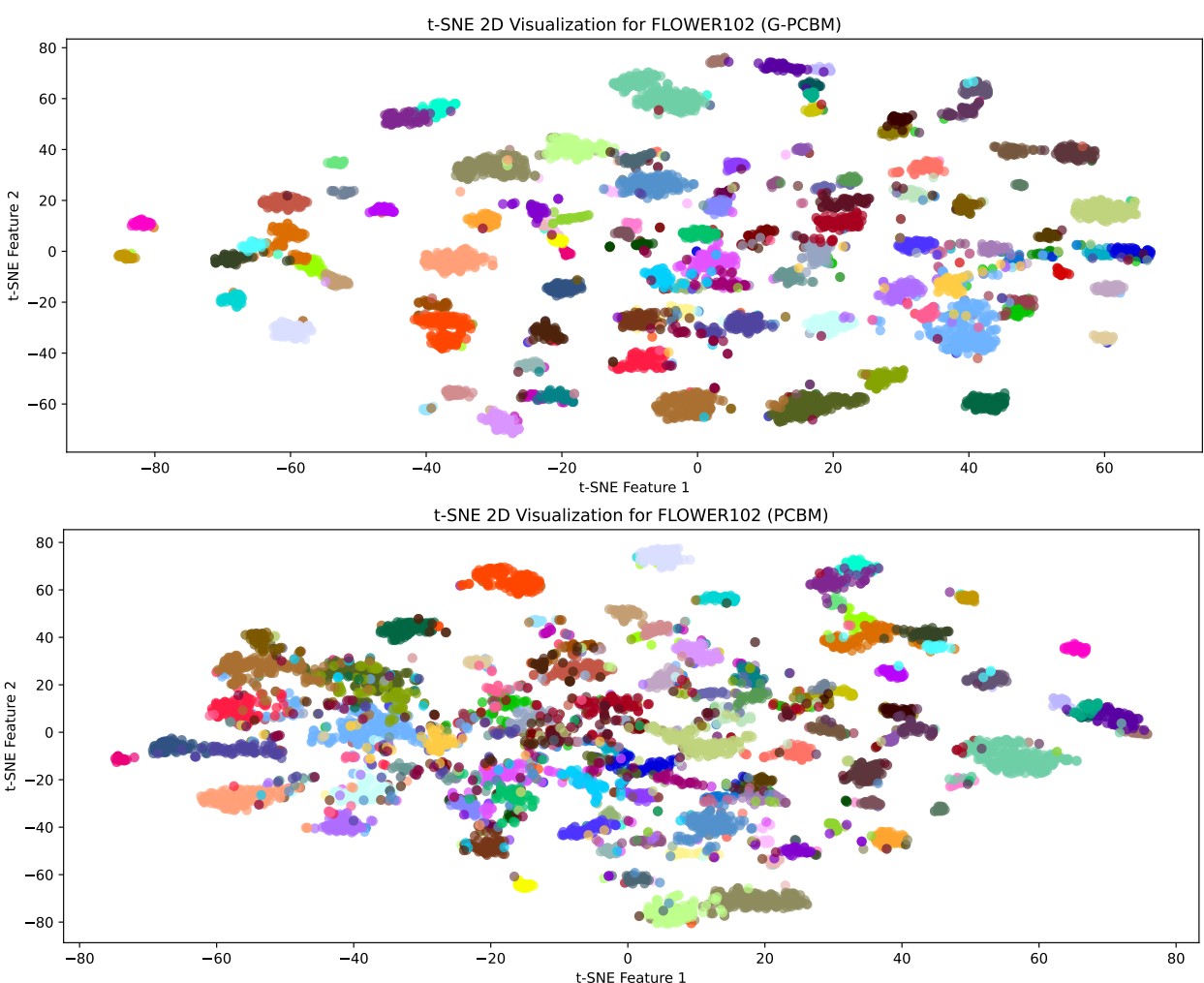

Figure 6: T-SNE visualization of concept activations distribution. *Top* shows a better clustering, while *bottom* has a more chaotic spread. Nodes with the same color are from the same label category.

We want to understand the essential benefit brought by the latent graph when the model needs to make predictions. In order to offer a visualized explanation and evidence, we choose to draw the T-SNE plots for models trained on the Flower102 dataset (as this dataset contains sufficient many distinct labels and has a relatively balanced sample distribution). As shown in Figure 6, G-PCBM more likely admits isolated and independent concept score vector clusters, differentiating from PCBM which mixes some label groups spread up. Consequently, G-PCBM succeeds in gaining more expressivity for concept activations and boosting prediction performance substantially.

# F   Latent Graphs Offer Robustness under Label-Free Settings

Figure 7: Attacking different types of concepts (nodes) in the CUB dataset can result in different scales of performance degradation.

In this section, we further investigate the effect of latent graphs on concepts (nodes), and we design a simple experiment to test such impacts. We first partition concepts into connected concepts (node degree $\geq 1$) and isolated concepts, and we also keep the whole concept set as a baseline group for random attacks; then, we will randomly mask or perturb the same number of concepts in these groups separately; lastly, we let our model predict those corrupted concepts. We choose the CUB as the testing dataset for this experiment[1].

As shown in Figure 7, attacking connected concepts does not harm the model performance as much as attacking isolated concepts or random concepts. We hypothesize that the latent graph structure provides much better robustness towards connected concepts, while the isolated ones cannot benefit from it. When masking or perturbing a concept, there will be information missing for the final prediction layer. Nevertheless, the latent graph structure can aggregate neighborhood information to recover the masked or perturbed concept so that the final prediction layer is still able to make a reasonable prediction.

# G   Performance on other larger-scale datasets

| Method | Places365 | | | ImageNet | | |
|---|---|---|---|---|---|---|
| | Acc | $\alpha$ | $\beta$ | Acc | $\alpha$ | $\beta$ |
| PCBM | 55.16% ($\pm$0.11%) | - | - | 77.49% ($\pm$0.10%) | - | - |
| Graph-PCBM | **55.25%** ($\pm$0.08%) | 0.1 | 0.17 | **78.48%** ($\pm$0.10%) | 0.1 | 0.17 |

Table 10: We report the average accuracies from 4 different random seed experiments along with the standard deviation.

In this section, we investigate the performance enhancement in prediction and intervention brought by latent graphs on large-scale datasets. We choose *Places365* (López-Cifuentes et al., 2020), a scene recognition dataset composed of 10 million images comprising 434 scene classes, and *ImageNet* (Deng et al., 2009) which contains 14,197,122 annotated images according to the WordNet hierarchy. We train G-PCBMs on these two datasets with 10 epochs, 1024 batch size, learning rate at 0.01, and Adam optimizer.

In Table 10, the G-PCBM has a latent graph to help capture the intrinsic concept correlation and improve label prediction in both large-scale datasets. We continue to present the positive impacts of involving a concept latent graph inside the model on *ImageNet* as a case study for intervention. Figure 8 also indicates

---

[1]We select checkpoints with a similar amount of connected concepts and isolated concepts.

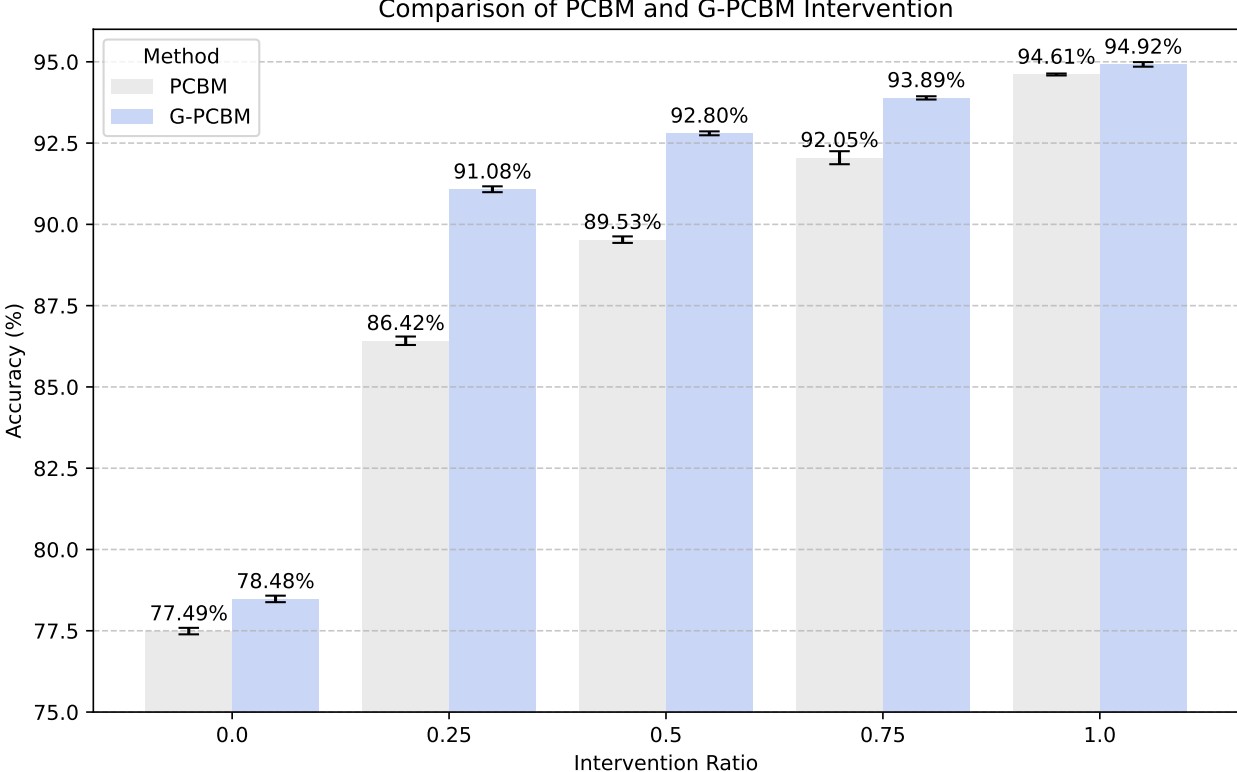

Figure 8: In a large-scale dataset, having latent graphs can still promote models to interact with intervention at different ratios.

the effectiveness of latent graphs in intervening large-scale datasets. In particular, G-PCBM significantly improves intervention performance in low intervention ratios. This also aligns with the results we conclude in the other datasets.

## H    Latent Graphs Benefit Various CBM Backbones

In this section, we validate the effectiveness of the latent graph on other CBM backbones. We choose CDM (Panousis et al., 2023) as the case study to show the benefits of latent graphs. The most important unit in CDM (Panousis et al., 2023) is the concept presence indicator which is modeled from a Bernoulli distribution, and the motivation behind the indicator variable is to sparsify the required concepts for label prediction. In Figure 9, applying latent graph to CDM can make a great enhancement in terms of label prediction, and latent graph can help to recover those filtered concept information to further boost model performance. The phenomenon is similar to what we have discussed in F, as masking attacks also sparsify concepts, and the latent graph can prevent the degradation caused by such attacks or operations. Along with the performance enhancements in Tables 1 and 2, learning latent graphs offers a robust generalization ability across different model architectures and training settings, which encourages the expectation of latent graphs effectiveness on other methodologies like LaBo, HardAR, and E-CBM.

## I    Salient Subgraph for Concept-Supervised Settings

The concept-supervised setting favors dense graphs, but it also easily leads to redundant connectivity for the latent graph, which will harm the concept graph interpretability and inference efficiency. We follow a heuristic strategy to find the salient subgraph inside the original concept graph: we mask one edge at a time and check the performance; if the performance remains the same or goes up, we delete that edge from the original subgraph. By doing this, we can improve our label prediction marginally without sacrificing concept

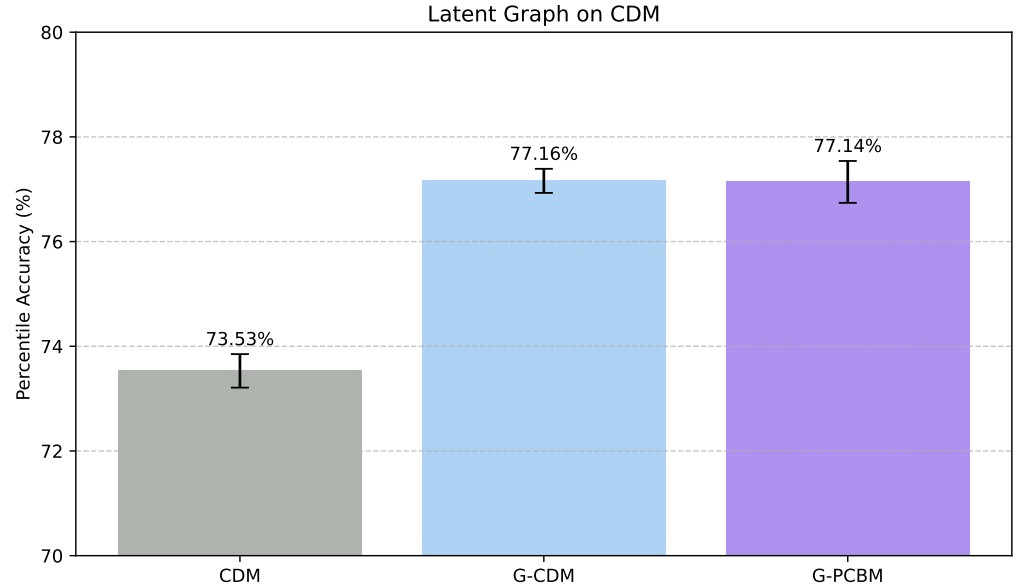

Figure 9: Adding a latent graph to the CDM can significantly improve model performance.

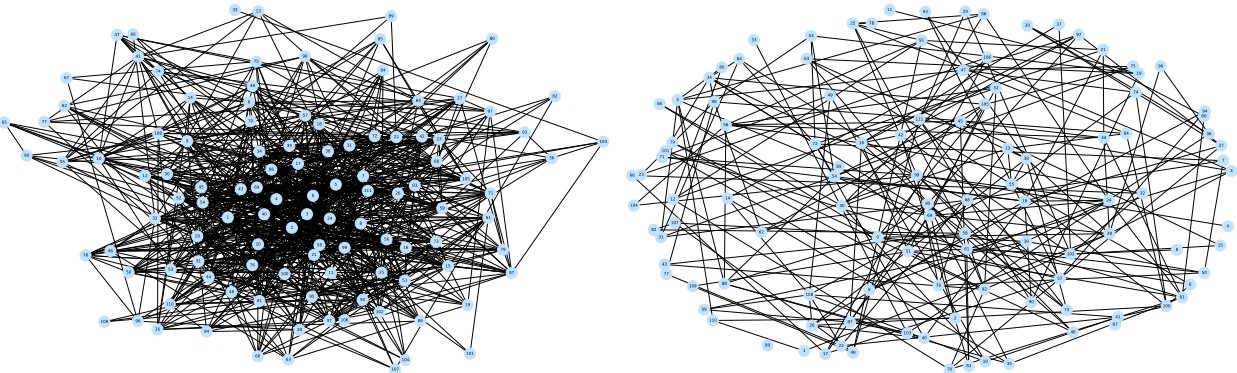

Figure 10: Both graphs are G-CBM concept graphs for the CUB dataset. Right: original concept graph. Left: salient subgraph structure.

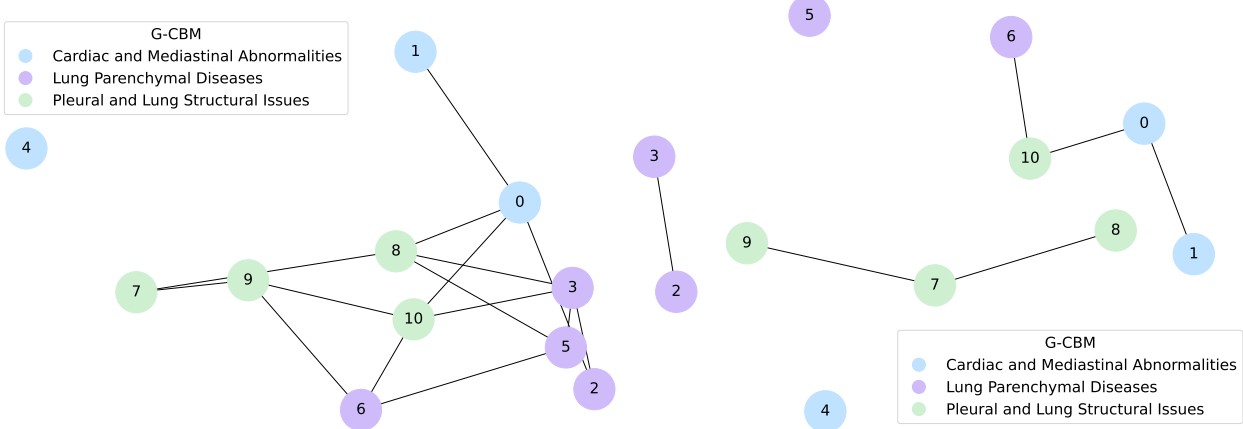

Figure 11: Both graphs are G-CBM concept graphs for the ChestXpert dataset. Right: original concept graph. Left: salient subgraph structure.

prediction (CUB label accuracy: 80.03% → **80.40%**; concept AUC: 83.20% → 83.20%; # of edges: 679 → **239**). Extracting the salient subgraph makes the concept graph more easily interpretable: for example in Figure 11, we can filter out the redundant edges and draw a more sparse concept graph. In the salient subgraph, concepts belonging to the same label class are more likely to connect. This also demonstrates that our latent graph effectively captures the hidden concept correlation embedded in the concept supervision.

We provide a quote from ChatGPT to show the reasonable connection between two node pairs: (node 6 (*Pneumonia*) and node 10 (*Pleural Other*)):

> A. Pneumonia can directly affect the pleura, leading to conditions like pleural effusion, empyema, or pleurisy. These pleural complications often arise due to inflammation or infection spreading from the lungs to the pleural space, making the relationship between pneumonia and pleural disease significant in both diagnosis and treatment.

And (node 0 (Enlarged Cardiomediastinum), node 10 (Pleural Other)):

> **Cardiac or vascular causes**: Enlarged cardiomediastinum often results from cardiac enlargement (e.g., heart failure or pericardial effusion) or vascular abnormalities (like aortic aneurysms). Some of these conditions can also cause pleural changes. For example:
>
> 1) Heart failure can lead to pleural effusion (fluid in the pleural space), which may manifest as a "Pleural Other" abnormality.
>
> 2) Aortic aneurysm or dissection may affect surrounding pleural structures due to proximity, causing pleural thickening or effusions.
>
> **Malignancies**: Tumors in the mediastinum (e.g., lymphomas or metastatic disease) can enlarge the cardiomediastinum and simultaneously invade or affect the pleura, leading to pleural abnormalities.
>
> **Infections and inflammatory conditions**: Severe infections like tuberculosis or mediastinitis can affect both the mediastinum and pleura, causing enlargement of the cardiomediastinum and pleural changes.

## J   G-CBM Intervention at full ratio

We compare our G-CBMs with E-CBMs (Xu et al., 2024) and standard CBMs (Koh et al., 2020) at full interventions. In Figure 12, G-CBMs can improve the performance significantly compared to the standard CBMs, and it can match up with E-CBMs as well. E-CBMs require energy calculations and gradient backpropagations to update the label probability, so the full intervention can offer sufficient information for E-CBMs to reach a good after-intervention performance. However, E-CBMs cannot handle low-ratio intervention well as shown in Figure 3 B, while our G-CBMs can be effective at different intervention ratios coherently. Plus, one-step intervention makes G-CBMs more efficient.

## K   Graph Complexity

**Graph CBMs favor sparse graphs under the label-free settings.** Without concept supervision, Graph CBMs prefer sparse graphs in general. As shown in Figure 13, the model gains performance improvement as we continue making the learnable graph sparse. Dense graphs result in over-smoothing concept scores, while sparse graphs can better preserve each concept value and propagate concept relation reasonably. We compare different graph complexity effects on the CUB dataset by varying the hyperparameter $\beta$ (large $\beta$ indicates more sparsity in the graph) under the Graph PCBM setting. We observe that our Graph PCBMs favor sparse graphs. The reason might be due to the way one collects concepts. If the model is trained under the label-free setting, one relies on a sophisticated concept generator like LLMs; however, LLMs will easily provide lots of redundant and repeated concepts. On the other hand, if the dataset contains concept annotations, concept sets are usually smaller in terms of the number of concepts and sufficiently informative. Moreover, concept

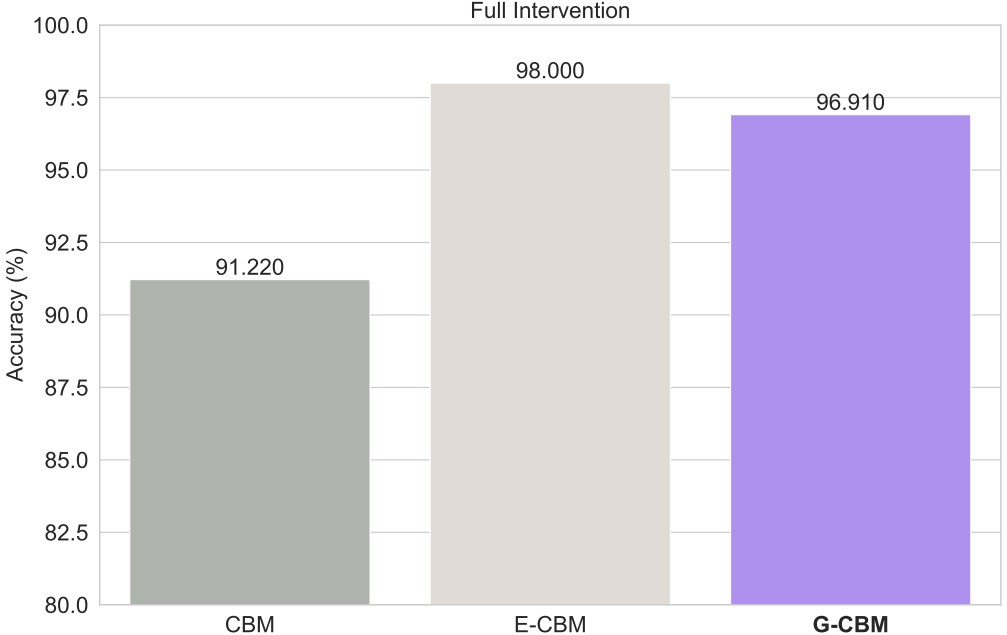

Figure 12: Comparison among models when full interventions.

annotations can act as an implicit concept correlation supervision, and models are expected to capture those hidden correlations in nature.

## L    Visualization of Learnable Graph

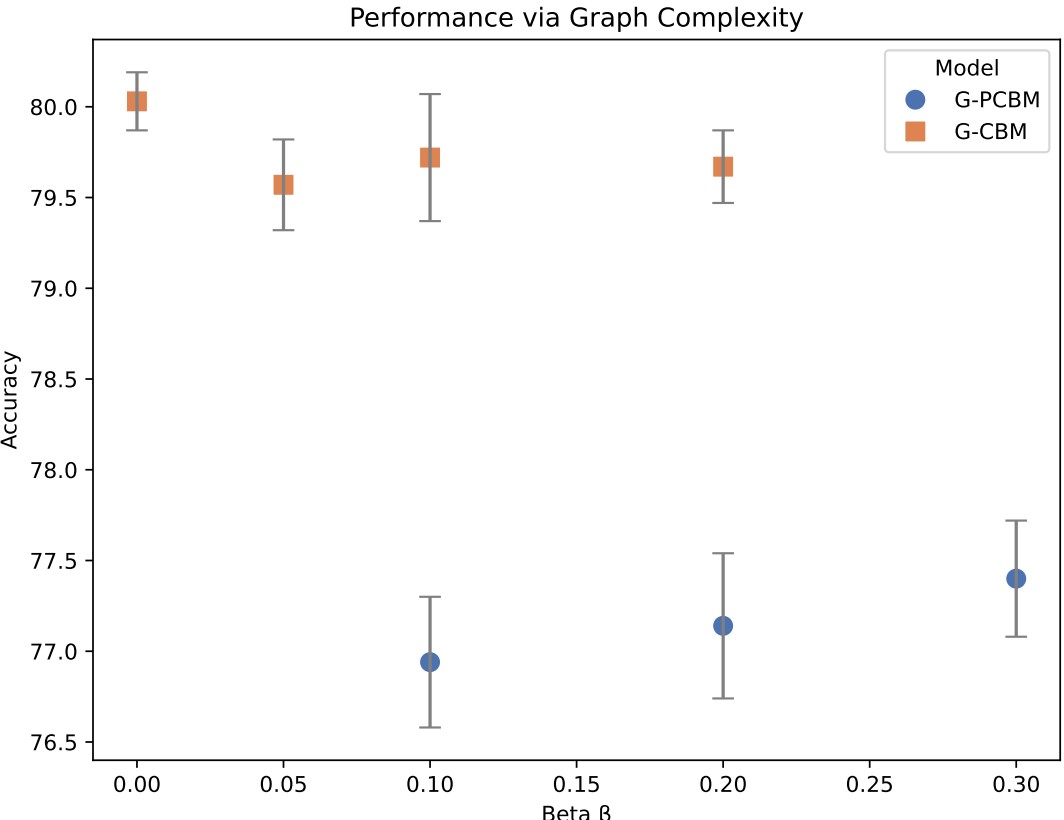

Figure 13: Performance changes as we set different $\beta$ value to control the latent graph complexity.

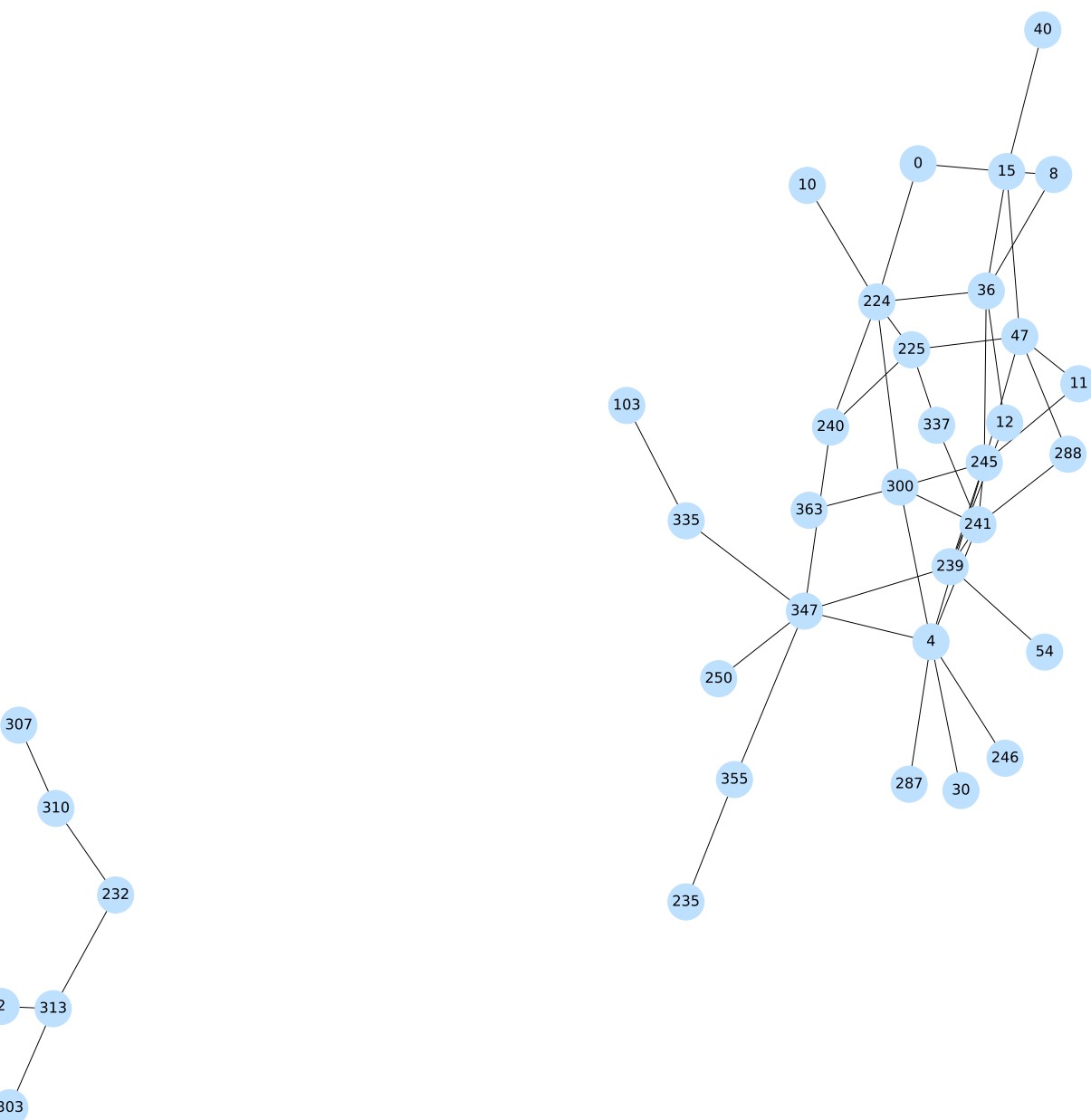

Figure 14: Zoom-in the densest component in the latent concept graph.

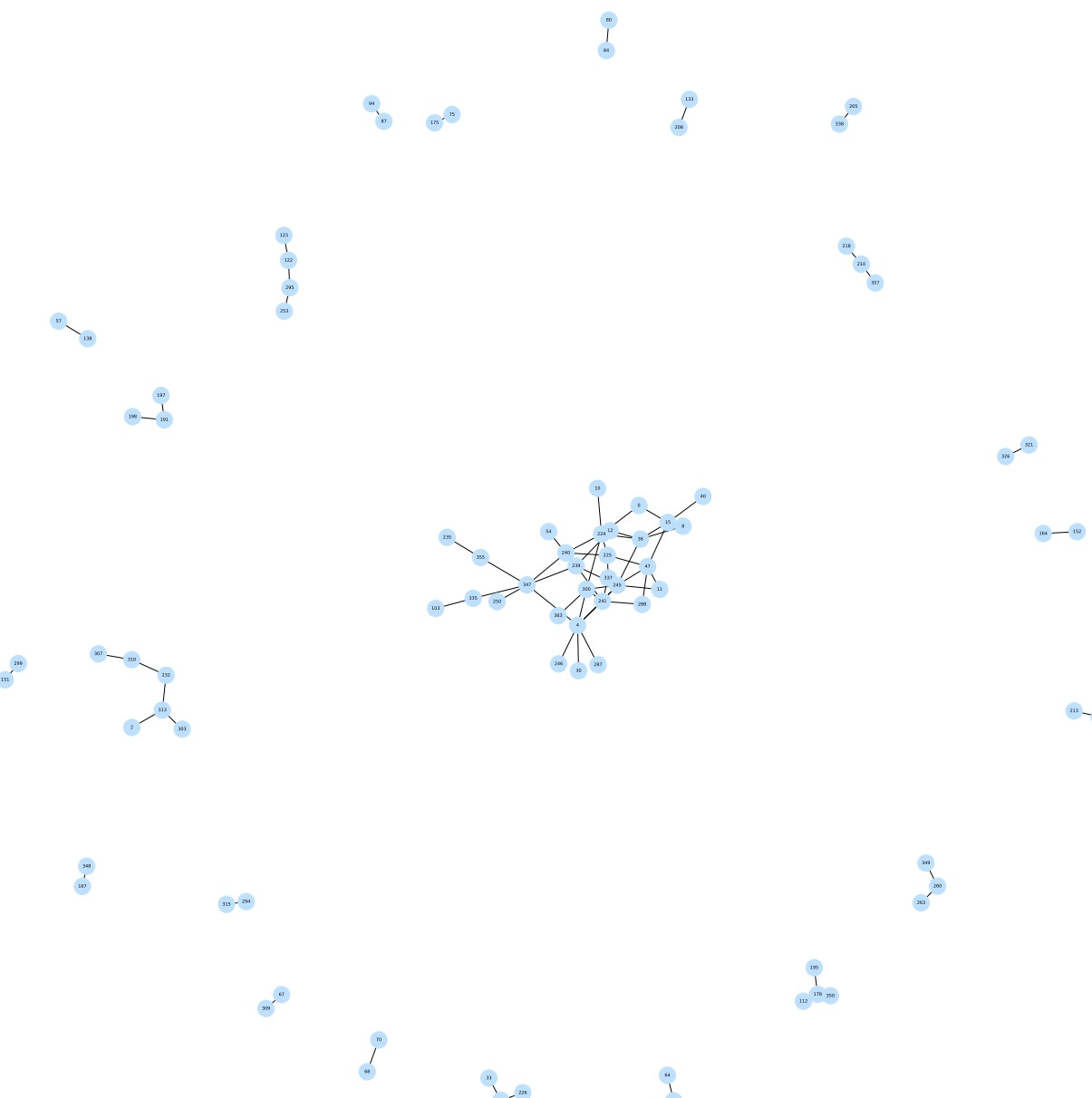

Figure 15: The overview of the CUB's concept graph in label-free settings.

