# OpenReview forum: "Graph Concept Bottleneck Models"
_TMLR — Accepted by TMLR_

### Review · Reviewer_2tRj · 2025-12-01

**Summary Of Contributions:**

The paper proposes an interesting variant of Concept Bottleneck Models (CBMs) that explicitly relates different concepts through a learnable graph structure, referred to as Graph CBM. The approach is flexible and can be incorporated into different methods, improving both performance and interpretability.

**Additional Comments:**

- The grouping of concepts, which promotes sparsity, seems to be key to the reported improvements in interpretability and performance. The use of a graph structure, along with the heuristic edge removal strategy described in Appendix I, is an elegant approach. However, the paper does not discuss alternative methods for inducing sparsity in CBMs. Could the authors comment on whether simpler approaches, such as weight pruning or auxiliary sparsity losses/regularization applied to a standard CBM, might achieve comparable performance gains? Including a discussion of such alternatives in the related work could strengthen the proposed approach.


- Interpretable models offer the opportunity to analyze performance under errors or hallucinations. Could the authors comment on this aspect? In particular, how does the model behave under out-of-distribution inputs? For example, if a cat image is presented to a model trained on the CUB dataset, would it be possible to clearly observe that the model cannot recognize the input, or would it still assign it to some of the bird classes/concepts?

**Audience:**

Yes

**Audience Explanation:**

Yes, the idea is very interesting, and improving model interpretability is an important and highly relevant topic.

**Broader Impact Concerns:**

None.

**Claims And Evidence:**

Yes

**Claims Explanation:**

The authors evaluate the proposed method on 8 different datasets (5 with unlabeled concepts and 3 with labeled concepts), including four different combinations of the graph structure with other methods. Overall, the proposed structure demonstrates improved performance.

**Requested Changes:**

Overall, the paper is well written. A small request:
- Figure 13 is difficult to read. Consider zooming in or reorganizing it similarly to Figure 4.

---

> ### Author Response · Authors · 2026-01-11
> **Official Rebuttal Response to Reviewer 2tRj**
>
> We sincerely thank Reviewer **2tRj** for the constructive feedback, and we are glad to learn that our submission is considered interesting, relevant, and impactful. We also appreciate the opportunity to discuss related topics raised by the reviewer, even when they extend beyond the primary focus of the paper.
>
> We first thank the reviewer for pointing out the readability issue in Figure 13. We have updated the figure in the revised draft by zooming in and reorganizing its layout for improved clarity.
>
> As noted by the reviewer, two interesting directions—concept sparsity in CBMs and hallucination/OOD behavior—were brought up. While both topics are valuable in their own right, we would like to clarify that **neither inducing sparsity in CBMs nor addressing hallucinations** constitutes the main problem we target in this work. Our primary objective is to enable CBMs to account for **intrinsic concept relations** and, by doing so, alleviate the accuracy–interpretability trade-off.
>
> ---
>
> ## 1. On Sparsity in CBMs and Our Design Choices
>
> The reviewer is correct that sparsity can contribute to interpretability. In our design, sparsity emerges mainly from (i) the salient graph construction and (ii) the $\ell_1$ penalty on edge weights. The motivations are twofold:
>
> 1. **Interpretability:** Removing superfluous edges makes concept connectivity clearer and improves human interpretability of the learned graph.
> 2. **Efficiency:** Sparse adjacency reduces message passing overhead and results in more efficient computation.
>
> For inducing sparsity directly within CBMs, we would like to highlight the LF-CBM baseline [1], which leverages a sparse linear predictor following the idea in [2]. As reported in [1], switching from a dense to a sparse prediction head can compromise model performance in certain cases. In our experience, this trade-off remains pronounced when model capacity is low—as is typical in CBMs—since predictive information may rely on multiple intermediate concepts.
>
> Given that CBM architectures are generally lightweight (in contrast to overparameterized deep encoders), the practical incentive for sparsity is less about compression and more about interpretability. In such cases, either a sparse prediction head or a sparse concept graph (as induced via $\ell_1$ normalization) is sufficient depending on the interpretability target.
>
> ---
>
> ## 2. On Hallucination and Out-of-Distribution Behavior
>
> To answer Reviewer 2tRj’s question regarding out-of-domain classification, the short answer is that the *model will **NOT** recognize OOD samples and will still predict one of the bird species*.
>
> We do not expect a CBM trained on the CUB dataset to generalize to unseen images such as cats. This is primarily because the CBM’s capacity and modeling ability are far below that required for broad generalization, unlike LLMs or other overparameterized models that benefit from large-scale pretraining and the double descent phenomenon. Although we utilize CLIP encoders to obtain representations for both images and textual concepts, the CBM module is always trained independently, and there is no direct parameter sharing between the encoder and the CBM. Therefore, while the CBM consumes outputs from an overparameterized encoder, this does not imply that the CBM inherits the encoder’s pretrained knowledge or can generalize to unseen datapoints.
>
> We agree with the reviewer that interpretability enables inspection of such failure modes. For example, we would expect OOD samples to activate concept vectors in an incoherent or low-confidence manner, which could potentially reveal model uncertainty or mismatch even without explicit OOD detection. However, this lies beyond the scope of the present work, and we view it as an interesting direction for future research.
>
> ---
>
>
> [1] Oikarinen, Tuomas, et al. "Label-free concept bottleneck models." arXiv preprint arXiv:2304.06129 (2023).
>
> [2] Wong, Eric, Shibani Santurkar, and Aleksander Madry. "Leveraging sparse linear layers for debuggable deep networks." International Conference on Machine Learning. PMLR, 2021.

---

### Review · Reviewer_vaNy · 2025-12-08

**Summary Of Contributions:**

The authors propose a method to extend concept bottleneck models (CBM) to not work with independent concepts but a concept graph. The method employs message passing along the edges in the context graphs and can be integrated in various CBMs. The authors report superior performance over conventional CBMs and present results on the robustness of graph CBMs.

Strengths:
- The method builds on the reasonable insight that often "concepts" are not independent
- The authors provide an extensive evaluation, spanning multiple aspects/perspectives

Weaknesses:
- The method still remains somewhat vague. For example, figure 1 does not comprehensively describe the method and the accompanying text does not clear up all the missing details. For example, how the concept graph is generated is unclear and requires a detailed description since it appears to be integral part of the approach.
- A lot of design choices seem somewhat arbitrary and are only accompanied by hand wavy explanations. For, example it is unclear why supervised approaches are susceptible to spurious correlations while unsupervised methods are not.

**Audience:**

No

**Audience Explanation:**

At the current stage the paper remains very vague and inaccurate s.t. I doubt that others would be greatly interested.

**Broader Impact Concerns:**

No.

**Claims And Evidence:**

No

**Claims Explanation:**

> To capture intrinsic interactions within the concept space, concept graphs must remain independent of label supervision, as labels can introduce spurious correlations that obscure genuine concept dependencies.

It is not clear why an unsupervised approach is not suffering from the same limitations

The "second contrastive" lacks justification and proper ablation. Just switching terms on and off (table 3) does not suffice since multiple losses require delicate balancing.

The claim
>  inherently permutationinvariant

is not well discussed explained.

While there is a slight outpeformance of unsupervised SOTA (with missing error bars), the proposed method does not outperform prior work under supervision, despite the claim in the abstract etc.

**Requested Changes:**

Rework method description. Due to the many parts, I suggest a self-contained main diagram detailing the model. Otherwise, address comments in other fields..

---

> ### Author Response · Authors · 2026-01-10
> **Official Rebuttal Comment to Reviewer vaNy**
>
> We appreciate the time and effort that Reviewer **vaNy** dedicated to evaluating our submission. We hope the clarifications below address the concerns and misunderstandings raised in the review.
>
> ---
>
> ## 1. Clarifying Methodological Details
> Thanks for pointing out the vague area. To address the reviewer’s concerns regarding the vagueness of the concept graph construction, we have revised the manuscript to explicitly detail the generation and learning phases. The key improvements are summarized as follows (and have been involved in the revised draft):
>
> The adjacency matrix $\mathcal{A}\in\mathbb{R}^{k\times k}$ is a learnable parameter that defines our concept graph. It may be initialized either (i) via similarity between image features and textual concept features or (ii) randomly (the standard practice). During training, $\mathcal{A}$ is optimized jointly with other model parameters and receives gradients from the task objective $\mathbf{CE}$ as well as the contrastive objectives $\mathcal{L}\_{\text{emb}}$ and $\mathcal{L}\_{\text{act}}$, following exactly the same optimization procedure as other neural parameters (e.g., MLP weights).
>
> We have integrated this explanation into Section 3.2.2 to make the initialization and learning procedure explicit.
>
> Regarding permutation invariance, our intention was to refer to a standard inductive bias of GNNs. Since this property is not the main focus of the paper, we now provide references for readers interested in the theoretical and architectural aspects of permutation invariance.
>
> ---
>
> ## 2. Intrinsic Concept Relations and (Un)Supervision
>
> We agree that the original sentence may be not clear enough. Our goal is to obtain supervision on the **intrinsic structure** among concepts rather than on the **mapping from concepts to labels**. Since datasets do not provide ground-truth relational information between concepts, we introduce contrastive objectives to serve as a form of self-supervision over concept interactions. We have removed the misleading sentence and replaced it with a clearer explanation along with the motivation for using self-supervision.
>
> ---
>
> ## 3. Justifying the Second Contrastive Term
>
> To elaborate on the role of the second contrastive objective, we present the following ablation on Graph-PCBM using the CUB dataset:
>
> | Models     | $\text{None}$       | $\mathcal{L}\_{\text{emb}}$ | $\mathcal{L}\_{\text{act}}$ | $\mathcal{L}\_{\text{emb}} + \mathcal{L} \_ {\text{act}}$ |
> |-|-|-|-|-|
> | Graph-PCBM | 76.89% ($\pm$0.65%) | 77.67% ($\pm$0.72%)         | 77.88% ($\pm$0.32%)         | **77.95%** ($\pm$0.69%)                                   |
>
> Regarding the concern about the necessity of the second contrastive term, we would like to clarify its motivation. The concept bottleneck introduces a projection from the high-dimensional image latent space to a lower-dimensional concept space, which leads to information loss. Empirically, directly classifying image features on CUB with a linear head yields 81.19%, while inserting the bottleneck reduces this capacity. The second contrastive loss mitigates this degradation by aligning the geometry of the concept space with that of the image feature space, improving downstream prediction. Our ablations further show that contrastive learning alone does not benefit standard PCBM, but becomes beneficial when combined with graph-based message passing, indicating that it addresses a concrete failure mode rather than being an ad-hoc addition.
>
>
> ---
>
> ## 4. On Underperforming Certain Prior Works
>
> We emphasize that our method is **orthogonal** to prior SOTA methods. Recent papers published at top venues have also been accepted without surpassing the current SOTA in accuracy, for example NeurIPS 2024 S-CBM [1] fails to beat ICLR 2024 E-CBM [2]. The performance difference between our model and the reported SOTA is approximately 0.3%, which is marginal.
>
> Furthermore, the TMLR Acceptance Criteria explicitly state:
>
> > *Crucially, it should not be used as a reason to reject work that isn't considered “significant” or “impactful” because it isn't achieving a new state-of-the-art on some benchmark.*
>
> We therefore respectfully submit that not beating a particular SOTA should not be a rejection criterion, especially given that we do not claim SOTA in the abstract. Lastly, we appreciate the reviewer pointing out the missing error bars in Figure 2; we have added them in the revision.
>
> ---
>
> ## 5. Interest to the TMLR Audience
>
> The reviewer concludes that the topic would not interest the TMLR audience, citing vagueness and arbitrariness but giving only one or two concrete examples. We respectfully disagree, as the other two reviewers explicitly stated that the topic is both interesting and relevant. We would welcome more specific and actionable feedback from reviewer vaNy so that we can address any remaining points of confusion.
>
> ---
>
> [1] arXiv:2406.19272
>
> [2] arXiv:2401.14142

---

### Review · Reviewer_1nzf · 2026-01-04

**Summary Of Contributions:**

This paper proposes graph-based concept-bottleneck models wherein the pretrained image embeddings are passed through a graph neural network to obtain a concept vector interpretable through textual concepts. The paper introduces a plug-and-play graph-based extension of CBMs and optimization objectives to learn concept-relationships and a final concept-vector. The key benefit of this approach is the ability to enable robust concept level interventions while retaining higher predictive accuracy.

**Additional Comments:**

NA.

**Audience:**

Yes

**Audience Explanation:**

Yes, multi-modal models are of popular interest and this paper proposes an improvement to concept-bottleneck models that enables robust intervention.

**Broader Impact Concerns:**

NA.

**Claims And Evidence:**

Yes

**Claims Explanation:**

Partially, no. While the results of the paper do support the claims, additional experiments/information may be necessary to attribute the improvements to graph-based concept-model.

**Requested Changes:**

1. The paper requires few changes for enhanced readability/clarity. Here are a few suggestions:
* On page-4: "For each image $v_i$, the initial concept score vector is computed via the ~~dot product~~ matrix-vector product"
* On page-4, could you please clarify the shape of adjacency matrix $\mathcal{A}$: is it $k$ or $k+1$?
* On page-5, *"The two contrastive losses operate at different layers and ..."*: as per Eq (2), it appears that the two losses are computed based on the node-embeddings at the last-layer. Please clarify.
2. Perhaps I missed it, but could you please describe the details of how Graph-CBM extends LF-CBM and PCBM? More specifically, I am curious to better understand the source of gains shown in Table 1. Are they coming from Graph-CBM having a higher capacity? Or, is it due to the losses introduced? Or, both?
3. Table 4 is good. However, for reference, it may be good to also show the classification performance based on just CE-loss (i.e., no contrastive-terms): both, with the GNN and with a simple linear-projector at the end of embeddings.

---

> ### Author Response · Authors · 2026-01-10
> **Official Rebuttal Response to Reviewer 1nzf**
>
> We thank Reviewer **1nzf** for the thoughtful and constructive feedback, and we are glad that the reviewer finds the paper interesting and relevant. Below, we address each of the requested clarifications and concerns raised by the reviewer, and we believe that the proposed revisions will further improve the clarity of the paper.
>
> ---
>
> ## 1. Clarifications on Notation and Presentation
>
> We have incorporated the reviewer’s suggested clarifications into the revised manuscript:
>
> 1. We replaced the phrase *“dot product”* with *“matrix–vector multiplication”* to more accurately describe the computation of initial concept scores.
> 2. We now explicitly indicate the shape of the learned adjacency matrix as
>    \[
>    $\mathcal{A} \in \mathbb{R}^{k \times k}$,
>    \]
>    where $k$ denotes the number of concepts. Since the adjacency encodes intrinsic relations between concepts, both dimensions correspond to concept indices.
> 3. In response to the comment regarding the sentence *“The two contrastive losses operate at different layers and …”*, we clarified where the two losses act in the model. Specifically, the embedding-level contrastive loss $\(\mathcal{L}\_{\text{emb}}\)$ is computed in the latent representation space $\(\mathbb{R}^d\)$, while the activation-level contrastive loss $\(\mathcal{L}\_{\text{act}}\)$ is computed in the concept activation space $\(\mathbb{R}^K\)$, after the mapping $\(f_3 : \mathbb{R}^d \rightarrow \mathbb{R}^K\)$. Thus, the two losses do not both operate on the final node embeddings, and they regulate different semantic levels.
>
> ---
>
> ## 2. Source of Performance Gains and Relation to LF-CBM/PCBM
>
> We combine the reviewer’s questions regarding attribution and the comparison with LF-CBM/PCBM, as they are closely connected.
>
> **Short answer.**
> The primary source of improvement arises from introducing message passing over concepts. In CBMs, concepts are traditionally treated as independent units. Graph-CBM instead allows concepts to interact through a learned graph structure, which we argue aligns better with the underlying semantic organization of many domains. For example, in fine-grained recognition tasks, certain concepts reinforce or suppress others, and modeling such dependencies improves both prediction and intervention behavior.
>
> To make this attribution more explicit, we provide an additional ablation under the label-free PCBM setting on the CUB dataset. We compare:
>
> - the effect of adding $\(\mathcal{L}\_{\text{emb}}\)$ and/or $\(\mathcal{L}\_{\text{act}}\)$ to **PCBM**, and
> - the same set of losses to **Graph-PCBM**,
>
> yielding the following results:
>
> | Models     | $\text{None}$       | $\mathcal{L}\_{\text{emb}}$ | $\mathcal{L}\_{\text{act}}$ | $\mathcal{L}\_{\text{emb}} + \mathcal{L} \_ {\text{act}}$ |
> |------------|---------------------|-----------------------------|-----------------------------|-----------------------------------------------------------|
> | PCBM       | 74.69% ($\pm$0.16%) | 71.62% ($\pm$0.67%)         | 72.02% ($\pm$0.78%)         | 72.62% ($\pm$0.73%)                                       |
> | Graph-PCBM | 76.89% ($\pm$0.65%) | 77.67% ($\pm$0.72%)         | 77.88% ($\pm$0.32%)         | **77.95%** ($\pm$0.69%)                                   |
>
> These results illustrate three points:
>
> 1. **Graph structure alone helps:** Graph-PCBM outperforms PCBM even without contrastive losses, supporting the role of structured concept interaction.
> 2. **Contrastive learning benefits Graph-PCBM but not PCBM:** When added to PCBM, the contrastive objectives degrade performance. When added to Graph-PCBM, they provide further gains.
> 3. **Capacity is not the main driver:** The GNN adds only $\(O(k^2)\)$ parameters and does not modify or fine-tune the image encoder; thus, improved performance is not due to increased representational capacity (and the adjacency matrices are sparse).
>
> For completeness, we note that implementing contrastive learning for PCBM requires nontrivial changes: we introduce a separate 3-layer MLP on the activation vectors and approximate the graph-level embedding using a weighted text embedding, since no message passing structure exists. The fact that such modifications do not yield improvements suggests that contrastive learning alone is insufficient, and that **message passing provides the structural prior necessary to make contrastive learning beneficial**.
>
> We hypothesize that the discrepancy stems from at least two factors:
> 1. PCBM activations tend to be dense and unstructured, causing contrastive supervision to inject redundant or misaligned information; and
> 2. Without a relational inductive bias, the contrastive objectives act as a perturbation to pretrained concept alignment.
>
> We agree that designing a contrastively regularized PCBM variant could be an interesting research direction, but note that it would constitute a separate contribution beyond the scope of this paper.
>
> ---

---

> > ### Comment · Reviewer_1nzf · 2026-02-09
> > **Thank you!**
> >
> > Dear authors,
> >
> > Thanks for providing detailed clarifications and additional experimental results.
> >
> > I agree with the analysis that graph structure helps but we cannot conclude that capacity is not the main driver solely based on the parameter count: rather, capacity is defined by the hypothesis class induced by the architecture and constraints, not by the raw number of free parameters.
> >
> > Regardless, the contribution of this work is that graph structure helps and these results support that: so, I will update my feedback to reflect that.
> >
> > Thank you,
> > Sincerely,
> > Reviewer 1nzf.

---

> ### Author Response · Authors · 2026-02-09
> **Thank you**
>
> Thank you for your thoughtful follow-up. We appreciate your point regarding model capacity and will revise the manuscript accordingly to avoid overclaiming. We are grateful for your recognition of the contribution and your constructive feedback.

---

### Author Response · Authors · 2026-01-13
**Summary on the Updates in Revision**

We thank the reviewers for their helpful feedback. Below we summarize the major updates made in the revised submission to improve clarity, completeness, and technical rigor.

**(1) Clarified graph construction and learning.**
In Section 3.2.2, we added a dedicated paragraph detailing both the initialization and training dynamics of the adjacency matrix, and we now explicitly specify its shape $\mathcal{A}\in\mathbb{R}^{k\times k}$. This addresses the concern regarding how concept graphs are instantiated and learned during training.

**(2) Refined motivation for contrastive objectives.**
In Section 3.3, we revised the narrative to emphasize that the lack of ground-truth concept relationships motivates the use of self-supervised contrastive objectives. We further distinguish $\mathcal{L}\_{\text{emb}}$ and $\mathcal{L}\_{\text{act}}$ and clarify that they operate at different layers and serve complementary purposes.

**(3) Additional ablations on contrastive losses and GNN contribution.**
In Section 4.5, we include new analyses and the new Figure 4 showing the effect of adding $\mathcal{L}\_{\text{emb}}$ and/or $\mathcal{L}\_{\text{act}}$ to both PCBM and Graph-PCBM. The results reinforce our conclusion that the performance improvements are primarily driven by message passing over concepts, rather than by model capacity or contrastive objectives alone.

**(4) Improved figure readability.**
We added a zoom-in version of the concept graph visualization (new Figure 14) to complement the previous Figure 15, following reviewer suggestions on legibility.

**(5) Added statistical reporting.**
All figures and tables now include standard deviations and/or error bars to strengthen empirical claims.

**(6) Added references on inductive biases.**
We added citations and brief discussion regarding inductive biases such as permutation-invariance in GNNs for readers interested in architectural properties.

We hope that these revisions meaningfully address the reviewers' concerns and improve clarity and accessibility of the work.

---

### Decision · Action_Editor_wrNR · 2026-02-20

**Recommendation:** Accept as is

**Audience:**

Yes

**Audience Explanation:**

After clarifying the details of the method all reviewers agreed that the paper is relevant to a significant part of the TMLR audience.

**Claims And Evidence:**

Yes

**Claims Explanation:**

Initially, reviewers acknowledged that the motivation is reasonable, the approach is elegant, and that the claim of mitigating the performance-interpretability trade-off is somewhat supported by the presented experiments. However, several issues with the presentation were pointed out, some of which might affect the interpretation of results:
* Reviewer 2tRj and vaNy suggested to improve figures by zooming in (Figure 13) to improve legibility, adding error bars (Figure 2) to get a better idea of relative model performance, and ensuring that Figure 1 is comprehensively describing the method either in the figure itself or by providing an explanation of missing details in the caption.
* some descriptions were considered hand wavy making the corresponding design choices seem somewhat arbitrary.
* The overall description of the method seemed vague to reviewers.
* Attribution of performance improvements to specific proposed changes was initially not convincing
* It was requested to be clear about the meaning of capacity (defined by hypothesis class rather than raw parameter count)

The authors addressed most of these issues and provided clarification of their motivation for several design choices. The authors also provided some additional experimental results which helped to highlight that message passing was responsible for the improvements rather than contrastive loss or capacity. Reviewer vaNy stated that while most concerns of their concerns had been addressed, the ablation was still somewhat shallow, as the loss components were just switched on and off, instead of exploring a larger cross product of loss coefficients. However, all reviewers agreed that the experiments are convincing and that the revision improved clarity and all lean towards accepting the submission.